# Mayaro virus pathogenesis and immunity in rhesus macaques

**Whitney C. Weber**[1,2], **Caralyn S. Labriola**[1,3], **Craig N. Kreklywich**[1], **Karina Ray**[4], **Nicole N. Haese**[1], **Takeshi F. Andoh**[1], **Michael Denton**[1], **Samuel Medica**[1,2], **Magdalene M. Streblow**[1], **Patricia P. Smith**[1], **Nobuyo Mizuno**[1], **Nina Frias**[1], **Miranda B. Fisher**[3], **Aaron M. Barber-Axthelm**[3], **Kimberly Chun**[3], **Samantha Uttke**[3], **Danika Whitcomb**[3], **Victor DeFilippis**[1], **Shauna Rakshe**[4], **Suzanne S. Fei**[4], **Michael K. Axthelm**[1,3], **Jeremy V. Smedley**[1,3], **Daniel N. Streblow**[1,3]*

1 Vaccine and Gene Therapy Institute, Oregon Health and Science University, Beaverton, Oregon, United States of America, 2 Department of Molecular Microbiology and Immunology, Oregon Health and Science University, Portland, Oregon, United States of America, 3 Division of Pathobiology and Immunology, Oregon National Primate Research Center, Beaverton, Oregon, United States of America, 4 Bioinformatics & Biostatistics Core, Oregon National Primate Research Center, Oregon Health & Science University, Portland, Oregon, United States of America

* streblow@ohsu.edu

**Data Availability Statement:** All of the data presented is within the manuscript and the raw data is provided as supplementary information.

**Funding:** This work was supported as an ONPRC Core Support Pilot Project (P51 OD011092)

## Abstract

Mayaro virus (MAYV) is a mosquito-transmitted alphavirus that causes debilitating and persistent arthritogenic disease. While MAYV was previously reported to infect non-human primates (NHP), characterization of MAYV pathogenesis is currently lacking. Therefore, in this study we characterized MAYV infection and immunity in rhesus macaques. To inform the selection of a viral strain for NHP experiments, we evaluated five MAYV strains in C57BL/6 mice and showed that MAYV strain BeAr505411 induced robust tissue dissemination and disease. Three male rhesus macaques were subcutaneously challenged with $10^5$ plaque-forming units of this strain into the arms. Peak plasma viremia occurred at 2 days post-infection (dpi). NHPs were taken to necropsy at 10 dpi to assess viral dissemination, which included the muscles and joints, lymphoid tissues, major organs, male reproductive tissues, as well as peripheral and central nervous system tissues. Histological examination demonstrated that MAYV infection was associated with appendicular joint and muscle inflammation as well as presence of perivascular inflammation in a wide variety of tissues. One animal developed a maculopapular rash and two NHP had viral RNA detected in upper torso skin samples, which was associated with the presence of perivascular and perifollicular lymphocytic aggregation. Analysis of longitudinal peripheral blood samples indicated a robust innate and adaptive immune activation, including the presence of anti-MAYV neutralizing antibodies with activity against related Una virus and chikungunya virus. Inflammatory cytokines and monocyte activation also peaked coincident with viremia, which was well supported by our transcriptomic analysis highlighting enrichment of interferon signaling and other antiviral processes at 2 days post MAYV infection. The rhesus macaque model of MAYV infection recapitulates many of the aspects of human infection and is poised to facilitate the evaluation of novel therapies and vaccines targeting this re-emerging virus.

awarded by the National Institutes of Health. WCW was supported by an NIH grant T32GM142619-01. The funders had no role in study design, data collection and analysis, decision to publish, or preparation of the manuscript.

**Competing interests:** The authors have declared that no competing interests exist.

## Author summary

Mayaro virus (MAYV) is an arbovirus capable of causing debilitating arthritis and myalgia in humans and the virus is currently circulating in Central and South America. With several factors supporting viral emergence, MAYV is a public health risk due to the lack of FDA-approved countermeasures. Although non-human primate (NHP) infection models are well established for chikungunya virus (CHIKV) and the equine encephalitic alphaviruses, there is currently no well-characterized NHP model of MAYV pathogenesis. With evidence of well-established mouse models of MAYV infection and a report from 1967 demonstrating that MAYV infection of NHPs in a laboratory setting was feasible, we aimed to further characterize MAYV infection in three rhesus macaques. Following precursor studies in mice to identify an optimal viral strain for NHP infection, we subcutaneously challenged rhesus macaques and characterized viral pathogenesis and immunity over the course of 10 days. Our study establishes a framework for future evaluation of MAYV-specific treatments in this relevant animal model.

## Introduction

Mayaro virus (MAYV) is a re-emerging arthritogenic alphavirus responsible for numerous outbreaks that are increasing in frequency in the tropical regions of Latin America and the Caribbean. In 1954, MAYV was isolated from forest workers in Mayaro County, Trinidad and Tobago, but the virus is now endemic to 14 countries of Central and South America [1–3]. Travel-associated infections have occurred in these endemic regions and reported for people returning to the United States and Europe [4]. MAYV is related to and co-circulates with chikungunya virus (CHIKV), which is the most prevalent alphavirus contributing to several large outbreaks over the last several decades in over 110 countries [5]. In 2022, there were 383,357 reported cases and 76 deaths caused by CHIKV with Brazil bearing the brunt of the public health burden (265,289 cases and 75 deaths) [6]. Brazil is also home to the largest number of MAYV outbreaks and is continually faced with the threat of other arboviral infectious outbreaks including dengue fever and Zika [7]. MAYV is primarily transmitted by *Haemagogus* sp. mosquitos dwelling in tropical forests, but experimental studies have shown other species to be capable of transmission [8–13]. Transmission is maintained in sylvatic transmission cycles by non-human primate (NHP) primary hosts and by rodent or other secondary hosts [14]. Although evidence of urban transmission of MAYV has not been identified, in research settings, MAYV has been shown to be transmitted by urban-dwelling mosquito vectors [1] including *Aedes albopictus* and *Aedes aegytpi*, causing concern for outbreaks outside of endemic regions [13,15–17]. While humans are only sporadically infected, some hypothesize that MAYV is poised to emerge more often due to tropical forest workers or travelers encountering more rural destinations [18–21]. Currently, there are no approved vaccines or therapeutics for the treatment or prevention of MAYV infections, presenting a major concern as MAYV continues to emerge in sporadic epidemics.

MAYV is an 11kb single-stranded, positive sense RNA member of the Semliki Forest antigenic complex that includes Una (UNAV), chikungunya (CHIKV), O'nyong'nyong (ONNV), Bebaru (BEBV), Getah (GETV), Semliki Forest (SFV), and Ross River (RRV) viruses [22]. Given the high degree of genetic and antigenic similarity within this serological complex, cross-reactive immune responses have been described for humans and in animal models [23–28]. There may be a high level of cross-reactive herd immunity afforded by CHIKV-MAYV

co-circulation, and cross-neutralization of MAYV by anti-CHIKV patient sera has been described by our group and others [23,24,29,30]. Phylogenetically, there are three distinct genotypic strains of MAYV (D, L, and N) with only 17% nucleotide divergence between them [10]. Genotype D viruses are distributed in Venezuela, Peru, and Bolivia, the L genotype is primarily confined to Brazil and Haiti, and Genotype N only contains isolates from Peru [31]. Due to co-circulation with other arboviruses, clinical disease similarity, and alphavirus cross-reactivity, these infections can also be difficult to diagnose as differentiating diagnostics are limited. There are incidences of arboviral co-infections, including reports of MAYV and CHIKV co-infection [32] and *ex vivo* superinfection with MAYV and Zika (ZIKV) [17]. Co-infections with non-arboviruses like HIV have also been reported, but little research has been done to investigate the interplay of these co-infections or consequence of pre-existing immunodeficiency [33]. Altogether, these confounding factors may lead to an underestimation of MAYV human disease burden.

MAYV causes Mayaro fever in humans which was first described in 1957, detailing case reports of febrile forest workers infected in 1954 and their MAYV-seroconversion [2,3]. Although disease is rarely fatal, it is estimated that 90% of MAYV infections are symptomatic and the incubation period is approximately 8 days [7]. Disease initially presents with a high fever that is concurrent with peak viremia at 1–2 days post-infection (dpi), and viremia has been reported to last at least 4 days [34]. Other disease symptoms include rash (inclusive of exanthema), headache, dizziness, retro-ocular pain, diarrhea, vomiting, inguinal lymphadenopathy, myalgia, and arthralgia [18]. These symptoms can last 5–7 days but myalgia and arthralgia can persist in >50% of patients for months to years following infection [34]. Acute phase infections can also present with mild leukopenia and thrombocytopenia [7]. Neurological complications associated with more severe cases and myocarditis has been reported following CHIKV infection, thus cardiac involvement has been hypothesized for other arthritogenic alphavirus infections including MAYV [35–41].

MAYV infection in mice has been used to characterize viral pathogenesis and also to evaluate MAYV-specific countermeasures. Mouse models of MAYV infection have been reported for C57BL/6, Balb/c, CD-1, AG129, Rag1-/-, and IFNαR-/- mice utilizing different strains of the virus including: $MAYV_{BeAr505411}$ [25], $MAYV_{BeH407}$ [28,42], $MAYV_{TRVL}$ [43–45], $MAYV_{IQT4235}$ [46] and $MAYV_{CH}$ [47,48]. Una virus (UNAV) is closely related to MAYV and has been used in a limited number of mouse infection studies [25]. To our knowledge, a comparison of MAYV strain pathogenicity in mice has not been published, however, the impact of genetic diversity on viral fitness was recently explored for three strains *in vitro* [49]. Vaccination strategies targeting MAYV have been reported for live-attenuated virus platforms [48,50], virus-like particles [51], adenovirus vectors [25,52], inactivated virus preparations [53], and DNA transfections [54]. Vaccines targeting CHIKV with cross-reactivity or cross-protective efficacy against MAYV have also been described [29,55]. Monoclonal antibody treatments [47,56,57] and antiviral drugs [58–63] directed against MAYV are also in development. Despite promising MAYV treatments reported in the literature, evaluation of their efficacy in NHP infection models has been hindered by the absence of an established NHP model.

NHP models of CHIKV infection have been well established in cynomolgus macaques (*Macaca fascicularis*) [64,65] and adult, aged, or pregnant rhesus macaques (*Macaca mulatta*) [66–69]. These models have proven useful for evaluation of CHIKV-specific vaccines [70–73] and monoclonal antibody therapies [74–76]. Additional NHP models of arthritogenic alphavirus disease have yet to be developed, although many have been established for the encephalitic alphaviruses. Indeed, Binn *et al.* established in 1967 that rhesus macaques could be infected with MAYV in a research setting, and the NHPs developed MAYV-neutralizing and CHIKV cross-neutralizing antibodies, which protected them from heterologous CHIKV

challenge [67]. However, this study has left several unanswered questions pertaining to viral tissue tropism, persistence, viral strain-specific differences in pathogenicity, as well as a general lack of knowledge about the kinetics and durability of innate and adaptive immunity. Due to the potential emergence of MAYV and the active and ongoing development of alphavirus-specific therapeutics and vaccines, we aimed to holistically characterize MAYV pathogenesis and immunity in adult rhesus macaques (RM).

## Results

### Infection of mice with the MAYV BeAr505411 strain results in robust replication and viral dissemination

To better inform strain selection for NHP experiments, we subcutaneously inoculated 4-week-old C57BL/6 mice (**Fig 1**) and 13-week-old IFNαR-/- mice (**S1 Fig**) in the right footpad with $10^4$ PFU of genotype D and L MAYV strains including $MAYV_{BeAr505411}$, $MAYV_{CH}$, $MAYV_{Guyane}$, $MAYV_{TRVL}$, $MAYV_{Uruma}$, or $UNAV_{MAC150}$ and compared viremia, tissue distribution and disease parameters for each of the strains. Infectious virus levels in serum collected at 2 days post-infection (dpi) from the female C57BL/6 mice were determined by limiting dilution plaque assays on Vero cells. Infection with $MAYV_{BeAr505411}$ and $MAYV_{CH}$ resulted in significantly higher serum viral titers compared to the three other MAYV strains tested (**Fig 1A**). Mice were euthanized at 5 dpi and MAYV vRNA levels were quantified using qRT-PCR for the RNA isolated from tissue homogenates of the contralateral and ipsilateral ankles, calves, and quads, as well as brain, spleen, and heart. Viral RNA levels generally trended significantly higher for $MAYV_{BeAr505411}$ with infection in muscles (ranging 1–3 logs higher) and joints (ranging 2–5 logs higher) compared to the other strains (**Fig 1B–1G**). Across viral strains the levels of viral RNA in ipsilateral joints and muscles were equivalent to the levels detected in the contralateral samples indicating efficient viral spread. In spleen, brain, and heart tissue homogenates, vRNA levels trended significantly higher (ranging 1–3 logs greater) for $MAYV_{BeAr505411}$ infection compared to the other MAYV strains (**Fig 1H–1J**). For many tissues, viral RNA levels in $MAYV_{CH}$ and $MAYV_{Guyane}$ infected C57BL/6 mice were similar to each other and higher than $MAYV_{TRVL}$, $MAYV_{Uruma}$, or $UNAV_{MAC150}$, but still lower relative to $MAYV_{BeAr505411}$ (**Fig 1A–1H**). Interestingly, the MAYV strain differences observed in C57BL/6 mice were not as profound in IFNαR-/- mice as the five MAYV strains all lead to similar changes in weight loss (**S1D Fig**) and footpad swelling (**S1C Fig**) as well as survival time (**S1B Fig**). However, in these immunodeficient mice UNAV infection exhibited the highest viral titer at 1 dpi as well as the quickest loss of body weight and time to death (**S1A, S1B and S1D Fig**). In summary, $MAYV_{BeAr505411}$ replicated to the highest levels in immunocompetent mouse tissues of expected viral tropism relative to other viral strains. Given these findings, we hypothesized that among the MAYV strains tested, $MAYV_{BeAr505411}$ would replicate the most efficiently in rhesus macaques and potentially elicit better clinical disease.

### Kinetics of MAYV replication in rhesus macaques reveals peak viremia at 2 dpi

To characterize MAYV pathogenesis in NHPs, we infected three male rhesus macaques (RM) ages 4, 10, and 13 years (**Fig 2**). At approximately one month prior to infection, we collected peripheral blood as well as spleen, axillary lymph node (LN) and mesenteric LN biopsies to provide baseline comparisons for immunological assays. Animals were inoculated subcutaneously in both hands and arms at five sites per arm (100μL per injection) in an attempt to mimic a mosquito bite with a total infectious dose of $1x10^5$ PFU of $MAYV_{BeAr505411}$. Peripheral

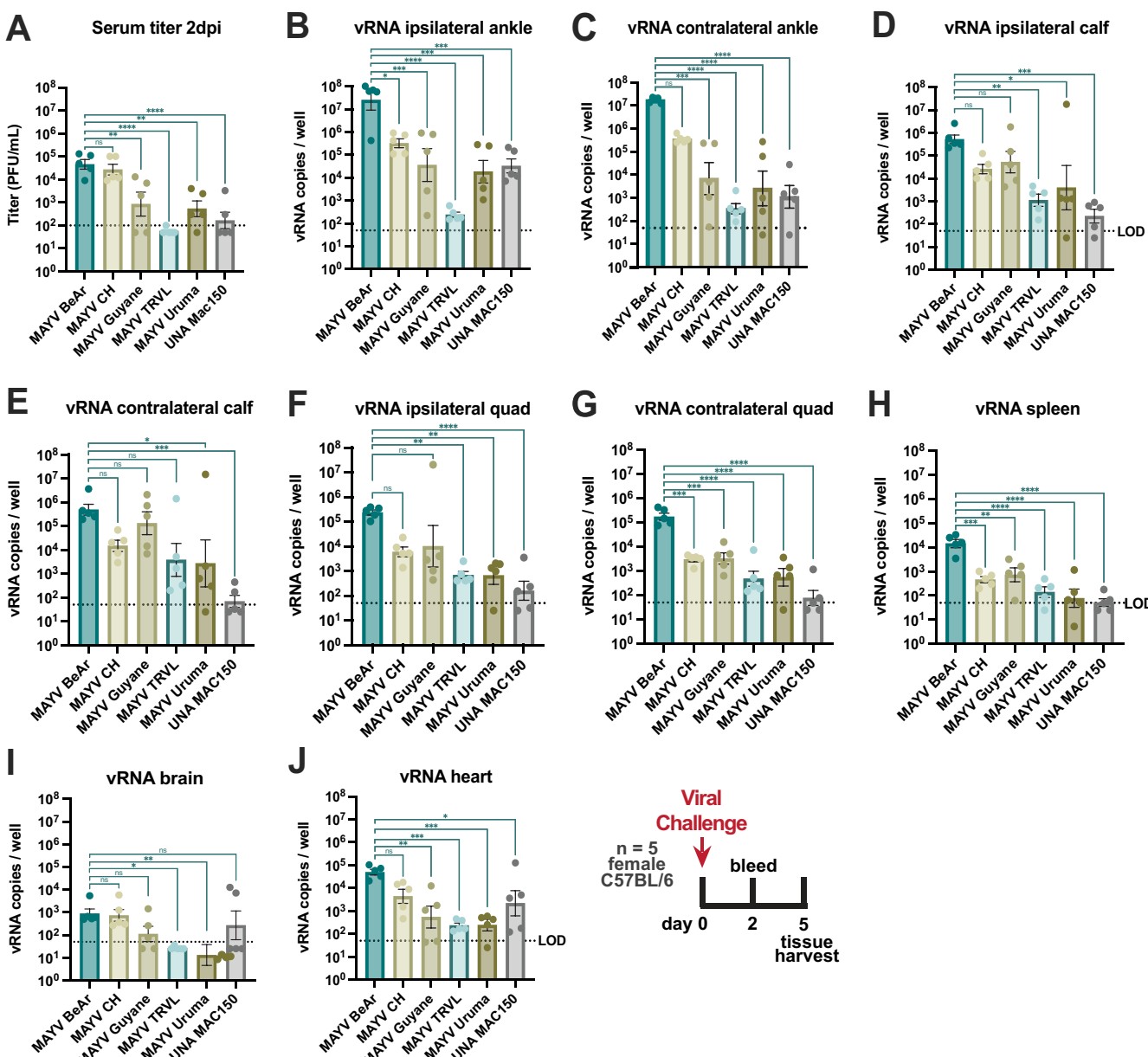

**Fig 1. Evaluation of MAYV strain pathogenesis in C57BL/6 mice.** Five 4-week-old female C57BL/6 mice were infected with $10^4$ plaque forming units (PFU) with one of five MAYV strains or UNAV via a right foot pad 20μL injection. Blood was collected for serum isolation at 2 days post-infection (dpi) and tissues were harvested at 5 dpi. Titers of infectious virus in serum at 2 dpi are shown in (**A**) and viral RNA (vRNA) levels in tissues were quantified (**B-J**). Data points are mean with SEM error bars for n = 5 per group, measuring three replicates of log-transformed data. Serum was tittered on Vero cells by limiting dilution plaque assays and vRNA in tissues was measured in triplicate by qRT-PCR (vRNA copies per well were normalized to the RsP17 house keeping gene.) The LOD in (**A**) was 100 PFU/mL with undetectable samples graphed as 50 PFU/mL. The LOD in (**B-J**) was 50 vRNA copies per well or per 200μL homogenate. Statistical analysis was completed using a one-way ANOVA with log-transformed data, where **** $p < 0.0001$, *** $p = 0.0001$, ** $p < 0.001$, * $p < 0.05$, ns $p > 0.05$.

blood and urine samples were collected at 0, 1, 2, 3, 4, 5, 7, and 10 dpi (**Fig 2A**). RM were humanely euthanized at 10 dpi for extensive tissue collection that included lymphoid tissues, muscles, joints, heart, peripheral nerves, central nervous system, male reproductive tissues, and other major organs. The 10 dpi timepoint was chosen to maximize the characterization of viral dissemination and immune activation following MAYV infection. We quantified plasma

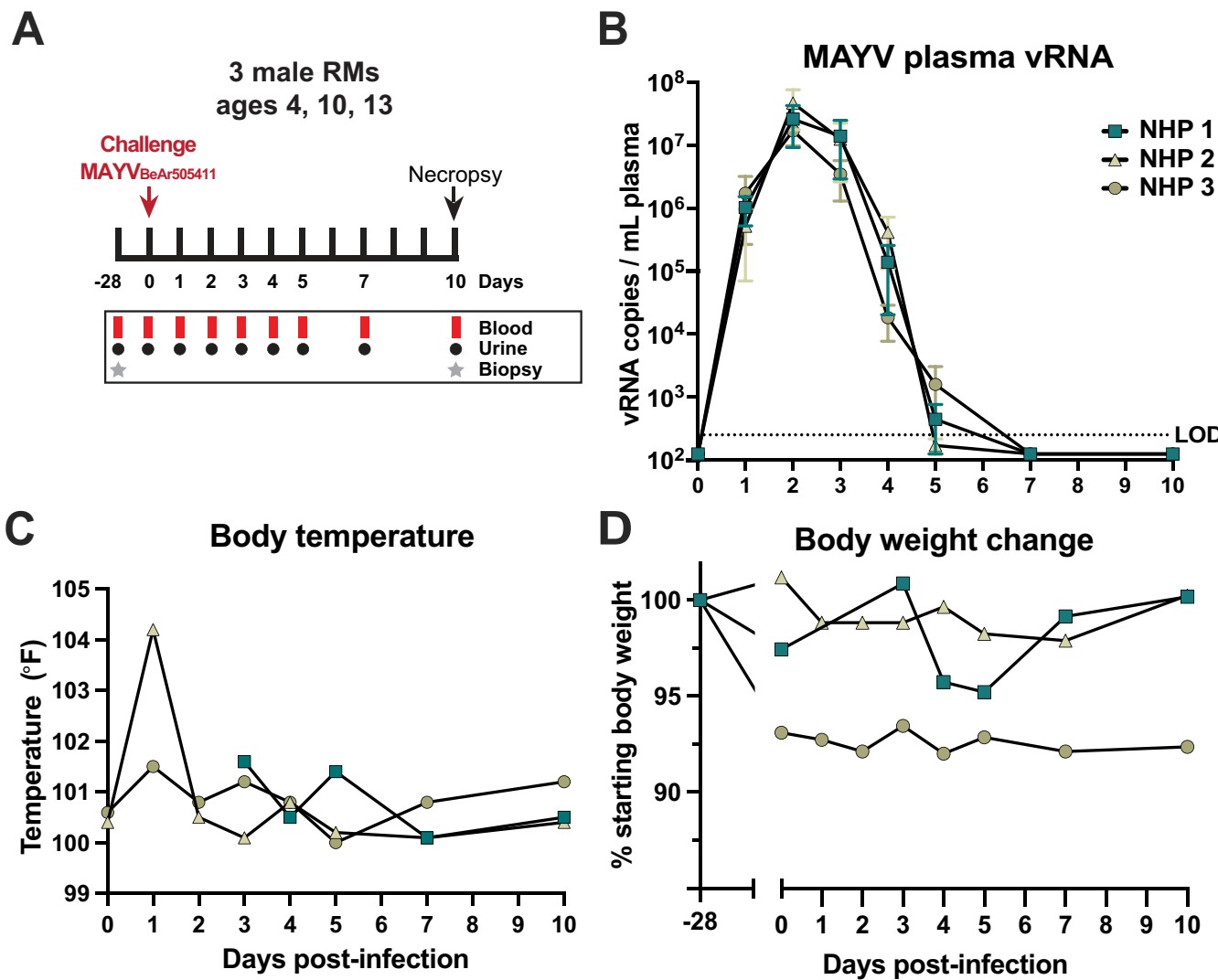

**Fig 2. Study overview of MAYV infection of NHPs.** Schematic summarizing the MAYV macaque infection study (**A**). Pre-infection axillary, inguinal, and mesenteric lymph node and spleen biopsies as well as blood were collected one month prior to infection for three male rhesus macaques (RMs) ages 4, 10, and 13 years. Animals were inoculated with $10^5$ plaque forming units (PFU) of $MAYV_{BeAr505411}$ administered subcutaneously and spread evenly in both arms and hands. Blood was drawn for PBMC and plasma isolation as well as complete blood count (CBC) and serum chemistry at 0–5, 7, and 10 dpi. Animals were humanely euthanized at 10 dpi and extensive lymphoid, muscles, joints, nerves, lobes of the brain, heart, major organs, and male reproductive tissues were harvested. Plasma was isolated from blood collections at 0–5, 7, and 10 dpi for quantification of viral RNA in copies/mL of plasma by qRT-PCR in triplicate reactions (**B**). The qRT-PCR data is representative of three independent experiments. The LOD was 250 copies MAYV RNA per mL of plasma with undetectable samples graphed as 125 copies vRNA/mL plasma. Body temperatures (˚F) (**C**) and change from starting body weight (%) (**D**) were recorded daily at all study timepoints.

viral RNA (vRNA) at all timepoints following infection and found that MAYV replicated up to $10^8$ vRNA copies / mL of plasma, with peak viremia occurring at 2 dpi in all three animals (**Fig 2B**). Plasma infectious virus was consistently detected at 1–4 dpi but not at 5, 7 or 10 dpi. (**Table 1**). We were unable to detect MAYV vRNA in urine samples from any of the RM, at any timepoint. Complete blood counts and serum chemistry analyses of each macaque revealed few remarkable changes over the duration of the study, but one animal experienced minor anemia that coincided with peak viremia (**S2 and S3 Figs**). One animal developed a fever of 104˚F at 1 dpi, however, the animals were only monitored for temperature during

**Table 1. Isolation of infectious MAYV from RM plasma and tissue.** Using NHP tissue homogenates collected in PBS at 10 dpi, we infected C6/36 cells and harvested supernatants at 3 dpi to isolate infectious MAYV. Viral supernatants were tittered in triplicate by limiting dilution plaque assays on Vero cells to quantify infectious viral particles in cell supernatants. Viral titers are reported as plaque forming units per 1 mL of C6/36 viral supernatant. The LOD was 3.3 PFU/mL. Samples with titers below the LOD are labeled (-).

| Plasma | NHP 1 | NHP 2 | NHP 3 |
|---|---|---|---|
| 1 dpi | $1.00 \times 10^{10}$ PFU | $4.60 \times 10^{10}$ PFU | $5.00 \times 10^{9}$ PFU |
| 2 dpi | $6.20 \times 10^{10}$ PFU | $4.66 \times 10^{10}$ PFU | $5.20 \times 10^{10}$ PFU |
| 3 dpi | $2.90 \times 10^{10}$ PFU | $9.25 \times 10^{10}$ PFU | $2.40 \times 10^{10}$ PFU |
| 4 dpi | - | $2.43 \times 10^{6}$ PFU | $3.40 \times 10^{4}$ PFU |
| 5 dpi | - | - | - |
| 7 dpi | - | - | - |
| 10 dpi | - | - | - |
| **Tissue** | | | |
| Ax LN | $1.00 \times 10^{2}$ PFU | - | - |
| Ing LN | - | $1.10 \times 10^{4}$ PFU | $3.30 \times 10^{0}$ PFU |
| Submandibular LN | - | - | - |
| Finger | - | - | - |
| Wrist | $1.43 \times 10^{6}$ PFU | - | - |
| Elbow | - | - | $3.10 \times 10^{6}$ PFU |
| Toe | $3.46 \times 10^{4}$ PFU | - | $1.36 \times 10^{4}$ PFU |
| Ankle | $1.96 \times 10^{4}$ PFU | - | - |
| Bicep | - | - | - |
| Brachial radius | - | - | - |
| Knee | - | - | $5.90 \times 10^{6}$ PFU |
| Quad | - | - | - |
| Tricep | - | - | - |
| Soleus | - | - | - |
| Hamstring | - | - | - |
| Aorta | - | - | - |
| Heart left atrium | - | - | - |
| Heart right atrium | - | - | - |
| Heart left ventricle | - | - | - |
| Heart right ventricle | - | - | - |
| Kidney | - | - | - |
| Liver | - | - | - |

procedures, making it impossible to know whether they were febrile at other times (**Fig 2C**). None of the three animals experienced weight loss over the duration of the 10-day infection study, although NHP 3 did experience 7% loss of body weight between the biopsy period and infection day. (**Fig 2D**). We did not observe additional signs of discomfort or disease in these three animals.

On the day of necropsy (10 dpi), a maculopapular rash was observed on the ventrum and flanks of one animal, without any observed pruritis, (**Fig 3A–3D** and **Table 2**) and these lesions were positive for MAYV RNA (**Fig 4E**). Erythematous macules, papules and xerotic plaques extended from the caudal thorax to the inguinal region with the most pronounced changes on the flanks. A bacterial culture revealed normal background dermatologic flora, and histologic screening for other etiologic causes such as measles virus was negative. Microscopic changes in the abdominal skin included multifocal acanthosis, mild dyskeratosis, and superficial edema which corresponded to grossly visible papules. Perivascular lymphocytic

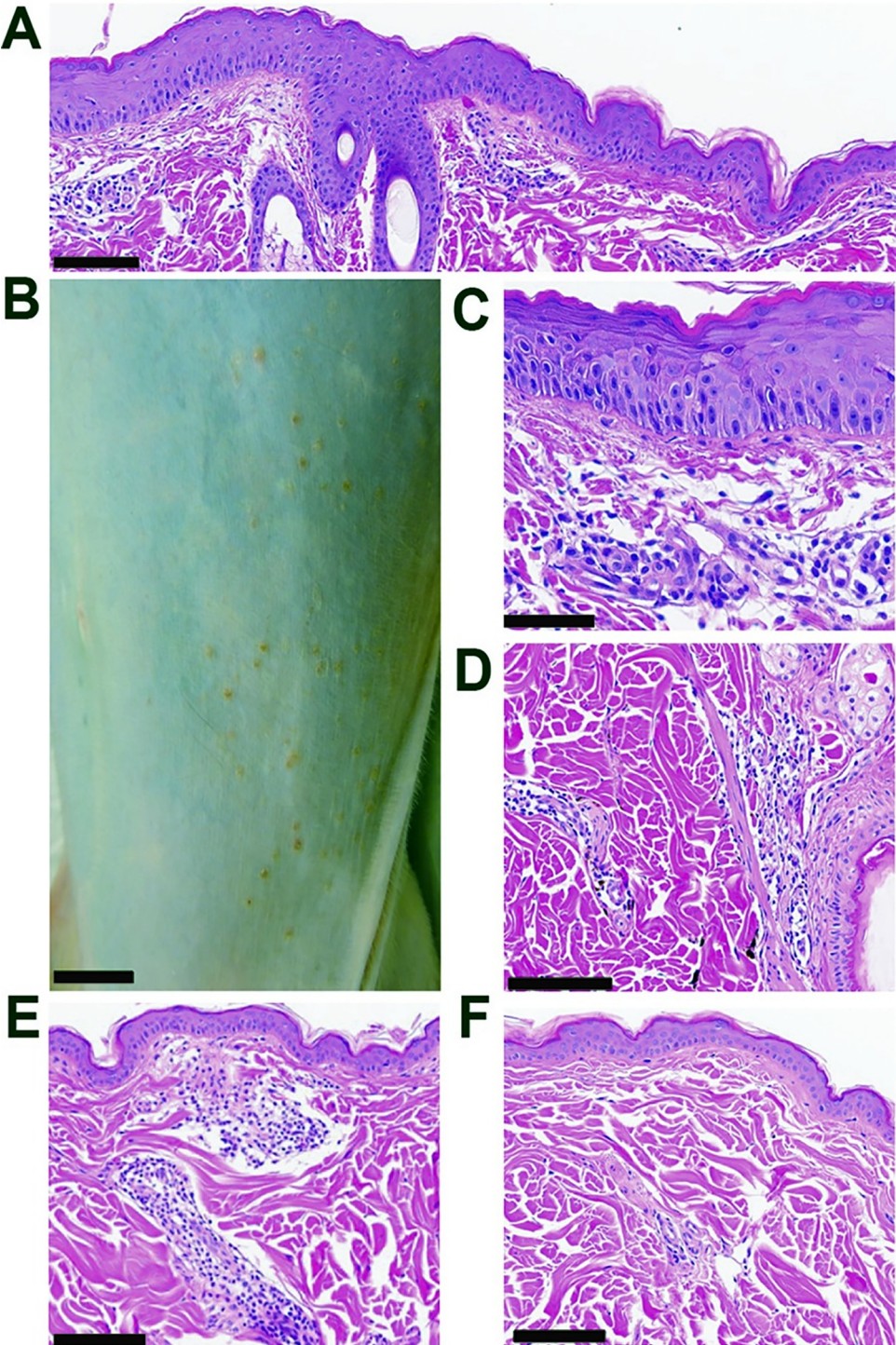

**Fig 3. Dermatologic pathology in MAYV-infected rhesus macaques.** At 10 dpi with MAYV, macaque skin sections were collected during necropsy, fixed, paraffin embedded, sectioned and stained for examination with hematoxylin and eosin (HE). (**B**; Bar = 1 cm) A maculopapular rash extends from the ventral abdomen to the flanks and inguinal region of NHP 3. (**A**; Bar = 100 μm. **C**; Bar = 50 μm) Sections of a maculopapular rash in the abdominal skin displaying multifocal acanthosis, mild dyskeratosis, superficial dermal edema, and perivascular lymphocytic inflammation in the superficial dermis. (**D**; Bar = 100 μm) The thoracic skin had similar perivascular and perifollicular lymphocytic aggregates. (**E**; Bar = 100 μm) Thoracic skin from NHP 1 with mild perivascular lymphocytic inflammation in the superficial dermis. (**F**; Bar = 100 μm) Normal thoracic skin from NHP 2.

**Table 2. Perivascular lymphocytic inflammation in the musculoskeletal, nervous, cardiovascular, and integumentary tissues of MAYV-infected rhesus macaques at 10 dpi.** Tissues are scored for presence of lymphocytic inflammation by relative intensity (+ to +++) or absence (-) of pathology within sections.

| Tissue | NHP 2 | NHP 3 | NHP 1 |
|---|---|---|---|
| **Joints** | | | |
| Elbow | - | ++ | - |
| Wrist/Fingers | + | +++ | + |
| Knee | ++ | + | - |
| Ankle/Toes | + | +++ | +++ |
| **Muscles** | | | |
| Biceps brachii | - | - | - |
| Triceps brachii | - | - | - |
| Brachioradialis | - | - | + |
| Quadriceps femoris | - | - * | - |
| Biceps femoris (hamstring) | + | - | - |
| Soleus | - | - | - |
| **Nervous Tissues** | | | |
| Cerebrum | + | + | - |
| Cerebellum/brainstem | - | + | - |
| Trigeminal nerve | - | - | N |
| Spinal cord / dorsal root ganglia | - | + (Lumbar) | + (Cervical) |
| Brachial plexus | - | + | - |
| Femoral nerve | - ** | - | - |
| Sciatic nerve | + | + | - |
| Eye | - | - | - |
| **Cardiovascular Tissues** | | | |
| Heart | + *** | - **** | + |
| Aorta | - | - **** | - |
| **Integument** | | | |
| Abdominal skin (rash) | N | +++ | N |
| Torso skin | - | +++ | ++ |

Tissues are scored for presence of lymphocytic inflammation by relative intensity (+ to +++) or absence (-) of pathology within sections using the following scale

+, one small aggregate of perivascular lymphocytes

++, multiple blood vessels within one or two areas of the tissue with small to moderate numbers of perivascular lymphocytes; and

+++, perivascular lymphocytes affecting a majority of blood vessels in small to moderate numbers with or without infiltration of the surrounding tissue.

N, tissue not available for evaluation.

*, minor myocyte degeneration and regeneration

**, pre-existing fasciitis

***, hypertrophic cardiomyopathy and valvular endocardiosis; and

****, myxofibromatous degeneration of the mitral valve (endocardiosis) and aorta.

inflammation was within the superficial dermis, which increased in severity in areas accompanying epidermal lesions. In the absence of gross or histologic epidermal changes, perivascular inflammation extended to the thoracic skin in this animal, as well as one other in the cohort, where it also surrounded few hair follicles (**Fig 3E**). One of the three animals did not have lesions within the thoracic skin sample that was collected, which was consistent with the negative viral detection as well (**Fig 4E**). Together, these data provide insight into the kinetics of MAYV viremia and disease symptoms.

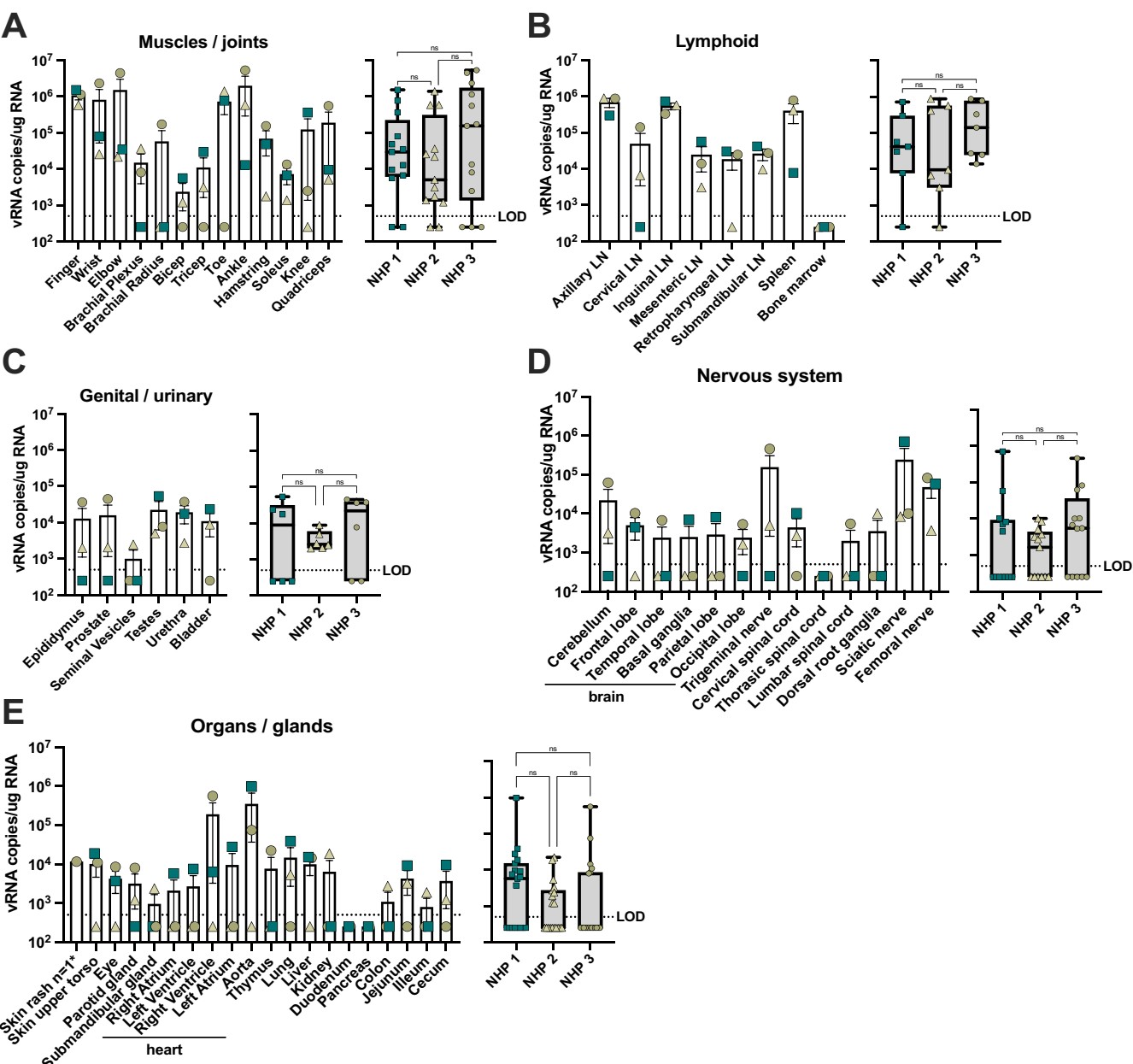

**Fig 4. Detection of MAYV RNA in NHP tissues at 10 dpi.** During necropsy extensive tissue subsets were collected from macaques and viral burden was determined through quantification of copies of viral RNA (vRNA) in qRT-PCR. Tissue subsets included muscles and joints (**A**), lymphoid tissues (**B**), genital and urinary (**C**), nervous system (**D**), and organs and glands (**E**). For all panels, the LOD was 500 copies per mL of tissue homogenate with undetectable samples graphed as 250 copies of vRNA/mL. Shown for each panel is a compilation and comparison of vRNA quantities for the tissue group, analyzed using a one-way ANOVA (ns = p > 0.05). All qRT-PCR reactions were performed in triplicate.

## MAYV infects joint, muscle, lymphoid, cardiac, and nervous system tissues of rhesus macaques

Next, we aimed to identify MAYV tissue distribution in the RMs at 10 dpi. Total RNA was isolated from muscle, joint, lymphoid, heart, brain, nerve, reproductive, and other major organs; and vRNA was quantified for each sample using qRT-PCR (**Fig 4**). At the time of necropsy, we combined right and left muscle and joint tissues into one sample tube and detected high levels

of vRNA in most subsets, notably the ankles, toes, elbows, fingers, and wrists for all three animals, indicating that the virus disseminated effectively throughout the body (**Fig 4A**). Consistent with this finding, we detected high levels of MAYV RNA for all three animals in several lymphoid tissues with the exception of bone marrow; viral loads were particularly high (nearly $10^6$ copies of vRNA per µg of RNA) in the axillary and inguinal lymph nodes (LNs), which drain from the arms and legs, respectively (**Fig 4B**). Viral RNA was detected in the male reproductive tissues (**Fig 4C**). MAYV crosses the blood-brain barrier in NHPs, as we observed vRNA in all three animals in lobes of the brain and other major central nervous tissues, the thoracic spinal cord being the only subset sampled with no detection in any animal (**Fig 4D**). Although we detected MAYV vRNA in many nervous system tissues, we did not observe evidence of neurological disease in any of the macaques. Because we detected viral replication in cardiac compartments in our mouse strain selection study (**Fig 1J**), we separated the ventricles, atriums, and aorta of the heart for viral detection in the RMs. We detected vRNA in all cardiac compartments for one animal, in the right ventricle of two animals, but one animal had no detectable vRNA in the heart tissue samples (**Fig 4E**). The duodenum and pancreas were the only tissues that were undetectable for vRNA for all three animals (**Fig 4E**). We compared the pooled vRNA levels in each tissue group across the three animals in an attempt to identify any trends in quantity or distribution, but there were no significant differences (**Fig 4A–4E**). The presence of infectious virus was determined by coculture of tissue homogenates with C6/36 cells and subsequent tittering of culture supernatants. Infectious virus was recovered in several muscle and joint tissues as well as lymph nodes, which provides additional evidence of sustained viral replication (**Table 1**). These data provide valuable insights into MAYV tissue tropism, replication, and distribution with valuable translational impact for understanding human infection.

## Immunopathologic changes associated with MAYV infection in rhesus macaques highlight variable tissue inflammation in joints, muscles, heart, and central nervous tissues

Histological assessment revealed that each of the animals infected with MAYV exhibited variable degrees of perivascular inflammatory cell infiltration in several tissue types. For example, all three animals had minimal to moderate lymphocytic inflammation of the finger, wrist, ankle, and toe joints (**Fig 5A and 5B**). The degree of inflammatory infiltration varied from minimally affecting rare perivascular areas in the fascia to moderate tenosynovitis also involving the adjacent adipose tissues (**Table 2**), and vasculitis was present in the most affected tissues. Multifocally, synovial and endothelial cells were hypertrophic, indicative of cellular activation. Interestingly, the ankles and toes (secondary sites of infection) of NHP 1 had more involvement than the forelimb joints. A focus of perivascular lymphocytes was in the brachioradialis muscle of NHP 1, which was the muscle collected closest to the infection sites. The elbow of one animal (**Fig 5C**) and the knees of two animals showed similar minimal to mild findings. Lymphocytic inflammation in the joint tissues occurred without gross changes in the cartilage or bone and variations were absent macroscopically where present on sections stained with hematoxylin and eosin (H&E). These findings imply that any pre-existing osteoarthritic components were less likely, though this cannot be ultimately ruled out due to collection limitations on size of tissue samples. Additionally, these lesions would be unexpected in the juvenile NHP 1.

Aggregates of lymphocytes were also present surrounding rare blood vessels in multiple additional tissues, including the appendicular muscles, heart, and nervous system of all NHPs (**Figs 5D–5G and S4, and Tables 2 and S1**). The medullary sinuses of the axillary lymph

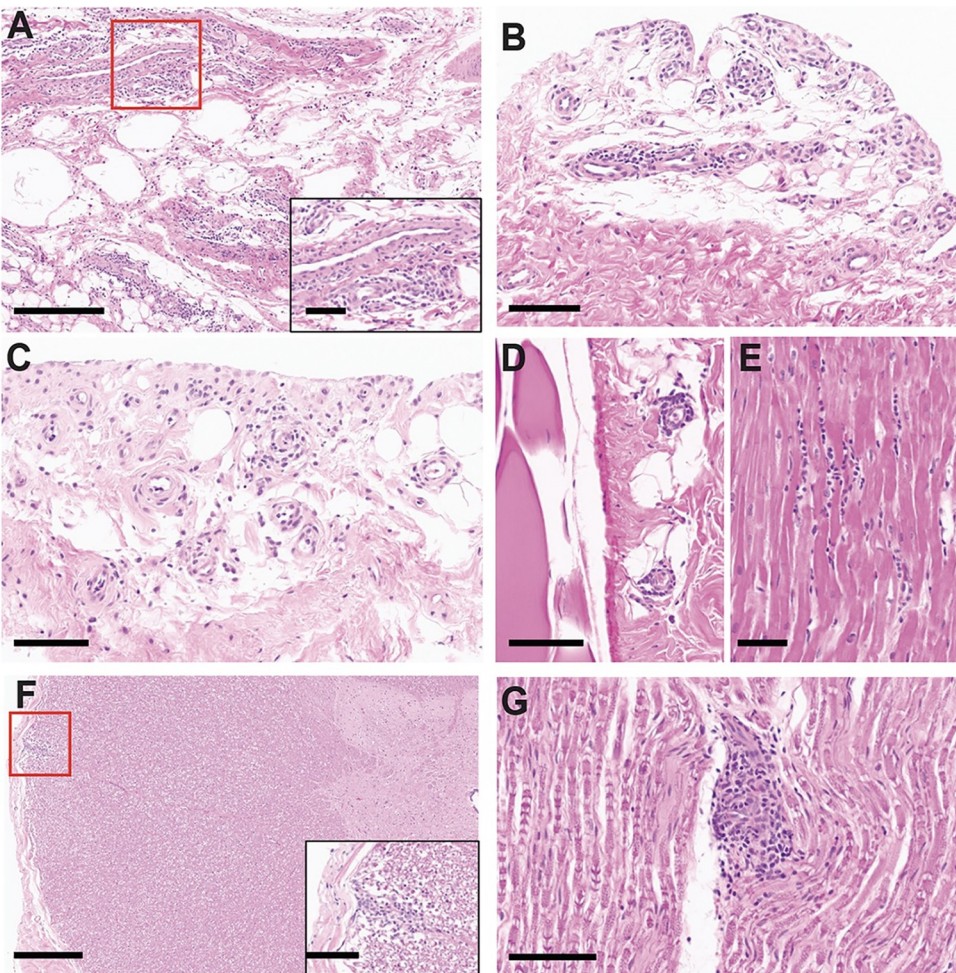

**Fig 5. Lymphocytic inflammation in the musculoskeletal, cardiac, and nervous system of MAYV-infected rhesus macaques.** Macaque joint and muscle tissues were collected during necropsy, fixed, paraffin embedded, sectioned and stained with hematoxylin and eosin (HE). Extensive histology was examined, and select representative images are shown for the three animals. (**A**; Bar = 300 μm, inset 100 μm) Lymphocytic inflammation within the periarticular connective tissue of the wrist and fingers with a perivascular pattern. (**B**; Bar = 100 μm) Perivascular and synovial lymphocytic inflammation in the ankle and toes. (**C**; Bar = 100 μm) Similar lymphocytic inflammation affects the elbow. (**D**; Bar = 100 μm) Minor perivascular inflammation within the fascia adjacent to the hamstring. (**E**; Bar = 50 μm) A minor focal aggregate of lymphocytes within the interventricular septum of the heart. (**F**; Bars = 500 μm, inset 100 μm) A vessel within the dorsal funiculus of the cervical spinal cord surrounded by lymphocytes. (**G**; Bar = 100 μm) Minor lymphocytic inflammation in the perivascular space of a vessel in the sciatic nerve.

nodes were expanded by histiocytes and hemophagocytes, which grossly presented as erythema and lymphadenopathy in all animals (S5 Fig and S2 Table). Mild enlargement of many peripheral and visceral lymph nodes microscopically corresponded to sinus histiocytosis and varying levels of hemophagocytosis, particularly present in the medullary sinuses of mesenteric and sacral lymph nodes. A consistent finding between these animals was lesions within the red pulp of the spleen (S5 Fig and S2 Table). At low magnification, a perifollicular pattern of congestion was evident (S5D Fig). At higher magnification, sinusoidal reticuloendothelial hyperplasia, histiocytosis, erythrophagocytosis, increased neutrophils, and rare microabscesses were evident (S5E–S5F Fig). Mentionable age-related or incidental lesions were chronic hepatic

degeneration and valvular endocardiosis of NHP 3 and hypertrophic cardiomyopathy, valvular endocardiosis, and fasciitis near the femoral artery and nerve of animal NHP 2.

## Cytokine and cellular innate immune signatures peak with MAYV viremia in rhesus macaques

We analyzed the expression of 37 cytokines and chemokines in longitudinal plasma samples following MAYV infection. Previously, a number of different proinflammatory cytokines and chemokines have been reported to be activated following MAYV infection in mice and humans or CHIKV infection in NHPs, in a process that typically coincides with viremia and subsequent innate and adaptive immune activation [69,74,77,78]. Studies with other arthritogenic alphavirus such as CHIKV and RRV have shown osteoblasts to be susceptible to infection, leading to secretion of MCP-1, IL-1, and IL-6 [79,80]. Consistent with these findings, G-CSF, IL-RA, eotaxin, MCP-1, IFN-α, and IFN-γ were all elevated relative to baseline at 2 dpi, aligning with peak viremia in the MAYV-infected RMs (**Figs 6 and S6**). Studies in mice following CHIKV infection have previously shown biphasic peaks in these inflammatory cytokines, and we captured sporadic secondary peaks for IL-4, IL-7, IL-8, IL-15, NGF-β, PDGP-BB, and SDF-1 (**S6 Fig**) [81]. Production of these cytokines and chemokines provide evidence of monocyte recruitment and migration (i.e., eotaxin, MCP-1) during peak viremia, which have a prominent role in the control of infection.

Activation of monocytes, macrophages, and dendritic cells have been consistently shown to contribute to the innate immune response to help control alphavirus infection but are also

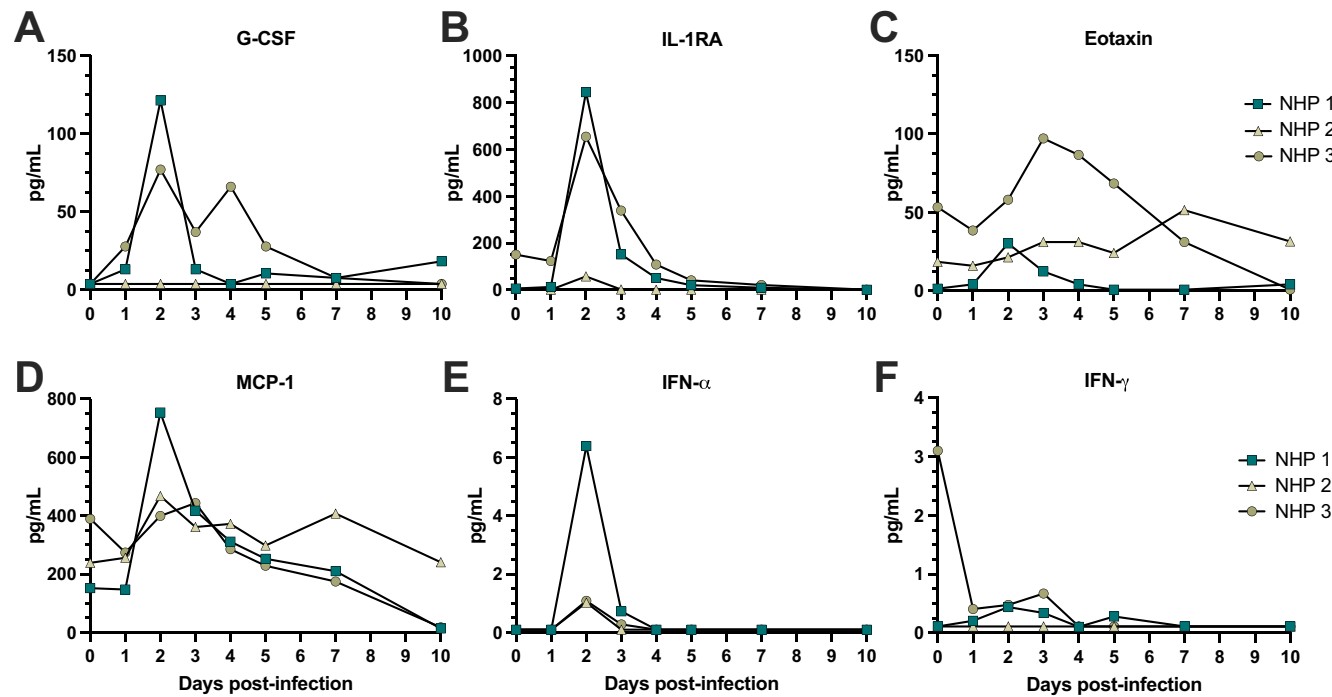

**Fig 6. Cytokine and chemokine profile following MAYV infection.** The inflammatory cytokine and chemokine profile following MAYV infection was characterized in macaque plasma at 0–5, 7, and 10 dpi using a Cytokine Monkey Magnetic 29-plex Panel for Luminex Platform Kit (Invitrogen) according to the manufacturer's instructions. A full panel of 29 cytokine and chemokine levels (pg/mL of plasma) were quantified, but shown are G-CSF (**A**), IL-1RA (**B**), eotaxin (**C**), MCP-1 (**D**), IFN-α (**E**), and IFN-γ (**F**). The LOD was determined to be the lowest detectable value in the assay for each cytokine or chemokine. Paired t tests were used for statistical analysis where baseline (d0) was compared to each of the other timepoints but did not yield any statistically significant results (p > 0.05).

capable of causing inflammatory damage [82–85]. To understand the kinetics of these innate immune responses in our MAYV-infected RMs, we quantified the frequency of total and activated (CD169+) monocytes, NK, and dendritic cells in longitudinal PBMC samples as well as lymphocytes isolated pre- and post-infection from lymphoid tissues (**Figs 7 and S7**). All three key peripheral blood monocyte populations (classical, non-classical, and intermediate monocytes) were highly activated in a process that coincided with the kinetics of plasma viremia (**Fig 2B**), peaking between 2 and 4 dpi but returning to baseline by 10 dpi (**Fig 7A–7C**). The peak of activation of NK cells (p = 0.3139), myeloid dendritic cells (p = 0.0460), and plasmacytoid dendritic cells (p = 0.0767) in PBMC also coincided with viremia (**Fig 7D–7F**), however, this trend was only statistically significant for myeloid dendritic cell activation (**Fig 7E**). While we detected increases in activation for monocyte, NK, and dendritic cell subsets, there were no significant changes in the total frequencies of any of these populations (**Fig 7A–7F**). Innate immune population activation in lymphoid tissues following infection varied by tissue and cell type. For example, after infection intermediate monocytes in the mesenteric LN were significantly activated (p = 0.0337) and those from the spleen also trended towards increased activation (p = 0.2798). However, other monocyte populations from these same tissues were not activated nor were they activated from axillary lymph node tissues (**Fig 7G–7I**). NK cell activation trended upwards following infection while not reaching statistical significance (p = 0.6650, p = 0.1481, p = 0.2022, respectively) (**Fig 7J–7L**). There was a general trend for plasmacytoid DCs to express less CD169 following infection and this trend reached significance in cells isolated from the spleen (p = 0.0264) (**Fig 7J–7L**).

## Proliferating T and B cell subsets dominate the early adaptive immune response to MAYV infection in rhesus macaques

The adaptive arm of the immune system is activated during alphavirus infection leading to the production of functional antibodies and T cells. While T cells have been shown to control alphavirus-mediated infection and disease [86–88], anti-CHIKV CD4+ T cells have also been shown in mice to mediate joint disease [88,89]. To characterize T cell frequency and phenotypic changes that occur in response to MAYV infection, we utilized flow cytometry for staining of longitudinal macaque PBMC from -28, 0–5, 7, and 10 dpi as well as lymphocyte preparations from lymph nodes and spleen collected at one month prior to infection and at 10 dpi (**Figs 8 and S8**). Using a well-characterized panel of antibodies, we found that the overall frequencies of each of the CD4 and CD8 T cell subsets remained stable in the peripheral blood with no major changes over time. While central memory (CM) CD4+ T cell proliferation (Ki67+) in peripheral blood increased at 2 dpi and again between 5 and 10 dpi (**Fig 8A**), only a slight increase in Ki67 staining was detected for the effector memory (EM) CD4+ T cells and less so for the naïve CD4+ T cell population. Proliferation of both CM and EM CD8+ T cell populations increased over time peaking at 7dpi (**Fig 8B**), which is consistent with previous published data for T cell proliferation in CHIKV-infected NHPs [66,74]. Also in line with published data, there was a steady expansion of granzyme B positive EM CD4+ and CD8+ T cells with peak frequency values attained at 7 to 10 dpi (**Fig 8C and 8D**) [90]. In addition, the frequency of granzyme B positive naïve and CM CD4+ and CD8+ T cells also increased following MAYV infection with peak values detected between 7 and 10 dpi, depending upon the specific subtype (**Fig 8C and 8D**). In general, CD8+ CM and EM T cells isolated from the spleen, axillary LNs, and mesenteric LNs stained significantly higher for the proliferation marker Ki67 and for granzyme B at 10 dpi when compared with tissues from prior to infection (**Fig 8E–8J**). While CD4+ T cell granzyme B was lower than those observed for the CD8+ cells, the CD4 + CM and EM T cell populations also changed their frequency of Ki67 and granzyme B

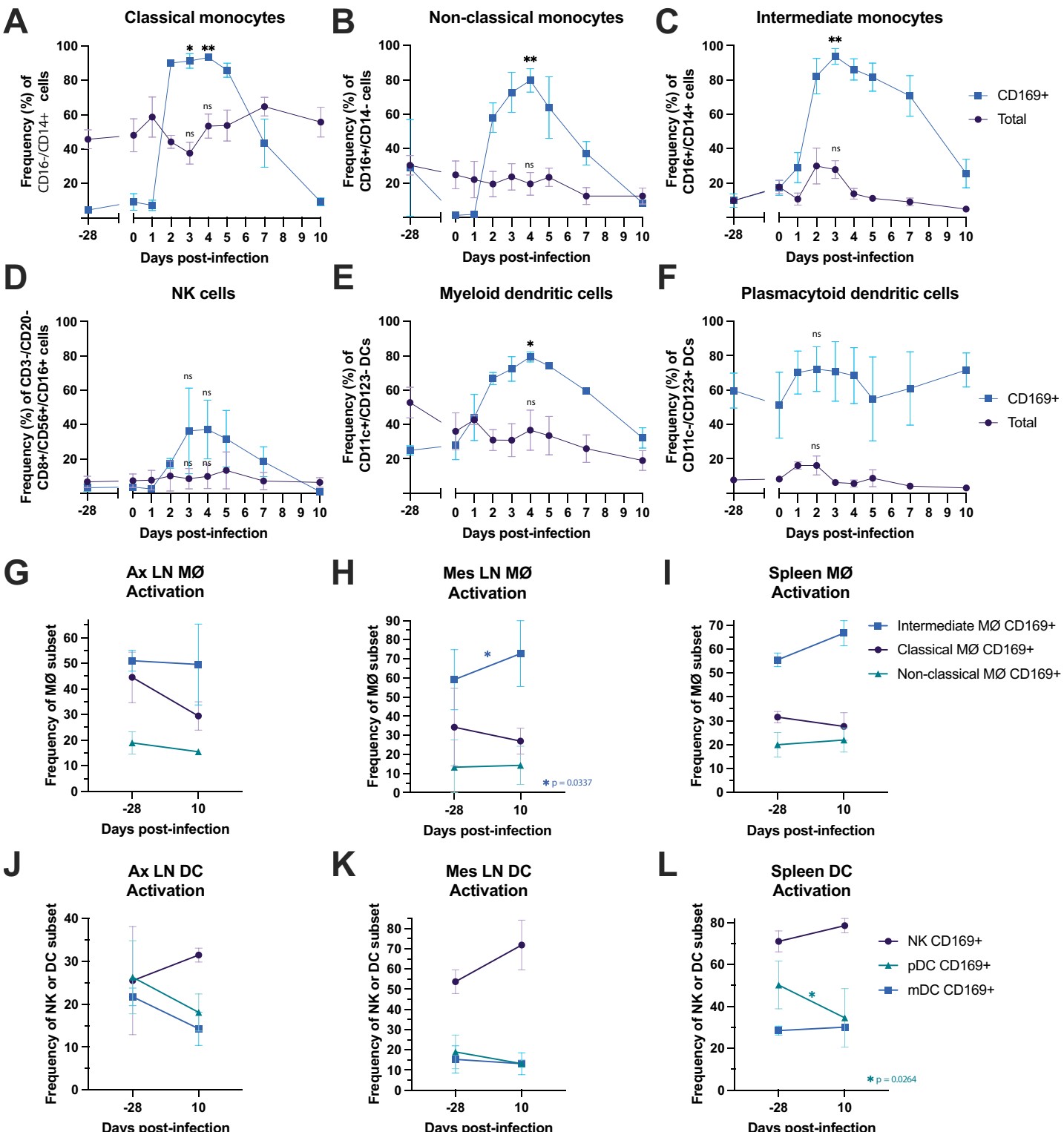

**Fig 7. Longitudinal peripheral blood and lymphoid tissue cell phenotype activation of monocytes, dendritic cells, and NK cells following MAYV infection.** Macaque PBMC from -28, 0–5, 7, and 10 dpi (**A-F**) and lymphocytes isolated from three lymphoid tissues either one month prior to infection or 10 dpi (**G-L**) were analyzed for cell phenotype using flow cytometry. Changes in the longitudinal frequency of both total and activated (CD169+) classical monocytes (**A**), non-classical monocytes (**B**), intermediate monocytes (**C**), NK cells (**D**), myeloid dendritic cells (**E**), and plasmacytoid dendritic cells (**F**) are quantified. Comparison of the frequencies of intermediate, classical, and non-classical monocyte phenotype activation at pre- or post-infection are quantified for axillary (Ax) LN (**G**), mesenteric (Mes) LN (**H**), and spleen tissues

(**I**). Frequency of NK cell and dendritic cell activation comparing pre- and post-infection is shown for axillary LN (**J**), mesenteric LN (**K**) and spleen tissues (**L**). Lines represent mean frequencies of the three animals and error bars represent the standard error of the mean. Longitudinal changes in total or activated (CD169+) cells in the peripheral blood (**A-F**) were analyzed using paired t tests where baseline (d0) was compared to the peak of the phenotype between 2 or 4 dpi; for this analysis, **** p < 0.0001, *** p = 0.0001, ** p < 0.001, * p < 0.05, ns p > 0.05. Statistical analyses for comparisons of baseline vs. 10 dpi cell phenotype frequencies in the lymphoid tissues (**G-L**) were completed using two-tailed paired t tests; only significant comparisons are shown, all other comparisons yielded ns p values > 0.05.

staining following infection but the responses were tissue and cell type specific with higher proliferation observed for cells derived from the axillary LN and spleen (**Fig 8E–8J**). Thus, these data demonstrate a robust cellular response following infection with MAYV.

Flow cytometry was also used to characterize the B cell component of the adaptive immune response by measuring the kinetics of B cell subset expansion and proliferation (Ki67+) in

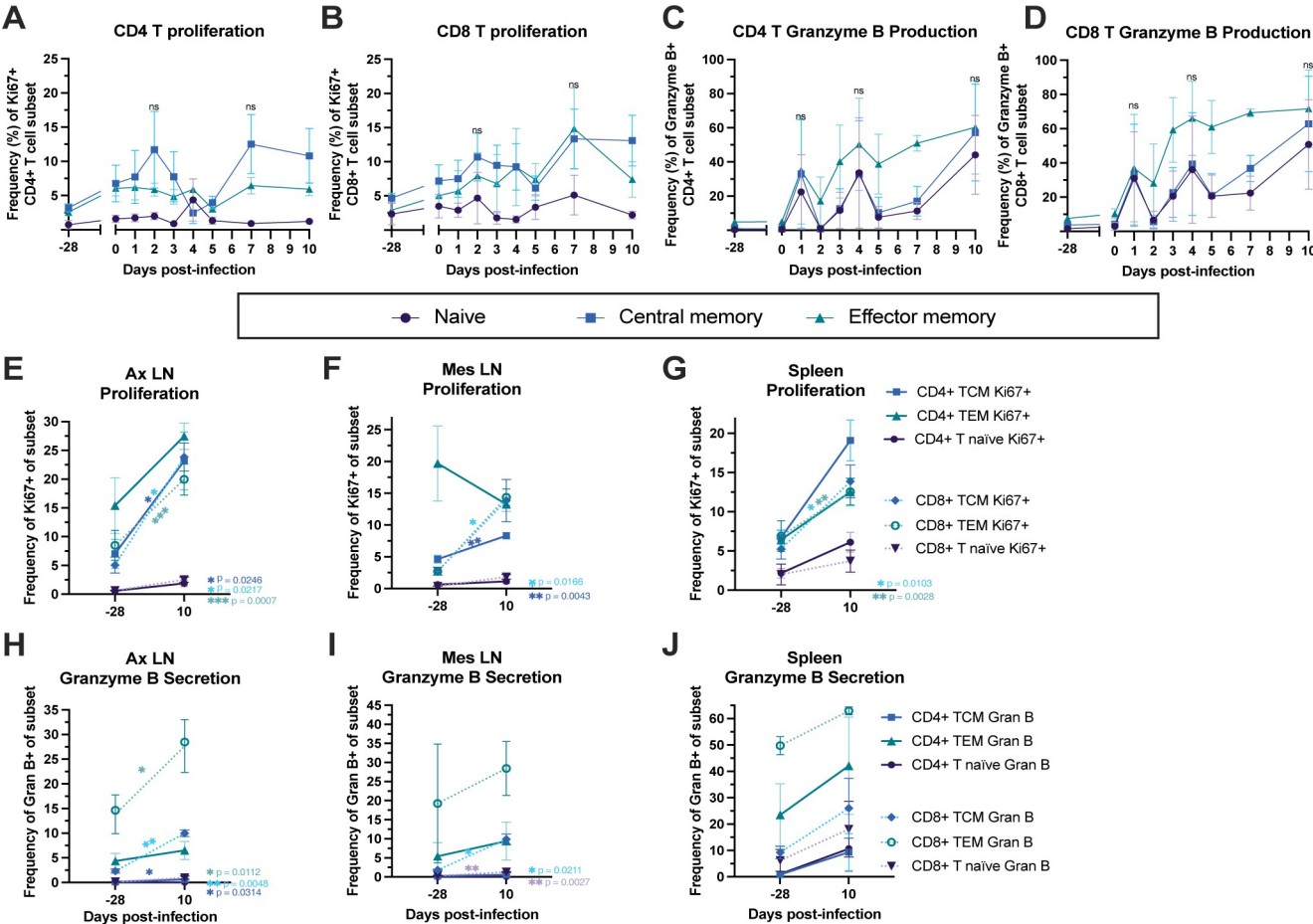

**Fig 8. Kinetics of T cell proliferation and granzyme B expression in peripheral blood and phenotype comparisons in lymphoid tissues pre- and post-MAYV infection.** Macaque PBMC from -28, 0–5, 7, and 10 dpi (**A-D**) and lymphocytes isolated from three lymphoid tissues either one month prior to infection or 10 dpi (**E-J**) were analyzed for T cell phenotype using flow cytometry. Changes in the longitudinal frequency of proliferating naïve, central memory, and effector memory CD4+ T (**A**) and CD8+ T cells (**B**) as well as granzyme B expression (granzyme B+) by CD4+ T (**C**) and CD8+ T cells (**D**) are shown. We additionally compared proliferation of these same memory T cell subsets at baseline to 10 dpi in the axillary LN (**E**), mesenteric LN (**F**), and spleen (**G**). We finally compared frequencies of granzyme B positive CD4 and CD8 memory T cell subsets in the axillary LN (**H**), mesenteric LN (**I**) and spleen from baseline to 10 dpi as well (**J**). Lines represent mean frequencies of the three animals and error bars represent the standard error of the mean. Longitudinal changes in proliferating (Ki67+) or granzyme B+ T cell subsets in the peripheral blood (**A-D**) were analyzed using paired t tests where baseline (d0) was compared to the peak of the phenotype at 2–3 timepoints; for this analysis, ns (not significant) represents p > 0.05 for naïve, central memory, and effector memory T cell subsets. Statistical analyses for comparison of baseline to 10 dpi cell frequencies in the lymphoid tissues (**E-J**) were completed using two-tailed paired t tests; only significant comparisons are shown, all other comparisons yielded ns p values > 0.05.

peripheral blood and lymphoid tissues over the infection time course (**S9 and S8** Figs). Similar to the T cell population frequency, no major changes in the total frequencies of naïve, marginal zone (MZ)-like, and memory B cell subsets were observed during the study period (**S9A Fig**) except for an expansion of proliferating MZ-like B cells that occurred between 5 and 10 dpi (**S9B Fig**). We did not detect major longitudinal changes in naïve or memory B cell proliferating subsets or proliferation of any B cell subsets in the axillary or mesenteric LNs (**S9B–S9D Fig**). However, we did identify an increase in proliferation of MZ-like B cells in axillary LN with a significant increase in cells from the spleen (**Figs 9E and S9C**). Proliferation of memory B cells trended higher in the axillary LN and spleen following infection but not in cells from the mesenteric LN (**S9C–S9E Fig**). These data suggest that MZ-like B cells are activated and proliferating following MAYV infection in the peripheral blood and spleen, likely for preparation of downstream differentiation into antibody-secreting plasmablasts.

To verify our immunology and pathogenesis findings at a global level, we performed RNA-seq analysis of longitudinal PBMC samples. Differential expression (DE) analysis revealed that several genes involved in interferon signaling (i.e., IFI6, IFI44, ISG15), antiviral immunity (i.e., STAT2, PARP14, MX1), and negative regulation of viral replication (i.e., OAS1-3, RSAD2, MX1) were significantly upregulated at 2 dpi (**Fig 9A**), with FDRp<0.05 and |FC|>2. If less stringent thresholds are used, other genes in these pathways such as IFIT1, IFNAR1, ISG15, and STAT1 are also differentially expressed (FDRp<0.2, |FC|>1.5). The top 10 enriched DEGs between 0 and 2 dpi, all key players in the antiviral response, were PARP12, SLC38A5, DTX3L, OAS1/3, STAT2, DHX58, DDX60, AGRN, and SIGLEC1 (FDRp<0.05, |FC|>2, ordered by FDR p-value) (**Fig 9B**). Similarly, Ingenuity Pathway Analysis (IPA) software identified changes in the enriched pathway signatures of both innate and adaptive immunity after MAYV infection. IPA also highlighted EIF2 signaling (translation modulation) to be tightly downregulated at 2 dpi while viral pathogenesis, interferon signaling, mTOR signaling, antiviral immune response, IL-12 signaling and production in macrophages, and B cell signaling pathways were among the top enriched upregulated pathways (FDRp<0.2, |FC|>1.5)(**Fig 9C and 9D**). These conclusions were well supported when examining these aspects for the 0 and 3 dpi comparison as well (**S10 Fig**). Over-representation of these innate and adaptive immune pathways support our findings and suggest an important role for the interferon response and antiviral immune responses following MAYV infection (**Figs 6, S6, 7, 8 and S9**).

## Virus-specific antibodies are present as early as 5 dpi and expand in neutralization breadth by 10 dpi

To interrogate humoral immune responses against MAYV, we measured the kinetics, magnitude, and breadth of antibody development following infection. Virus-specific IgM antibodies are typically present as early as 4 days post-infection but can persist for three months in humans [91–93]. In mice, evidence shows that CHIKV-specific IgM can be detected in serum as early as 2 dpi and CHIKV-specific IgG as early as 6 dpi, with both IgM and IgG anti-CHIKV antibodies having neutralizing abilities [94]. Consistent with these observations, we detected MAYV-specific IgM as well as IgG antibodies as early as 5 dpi in all three macaques (**Fig 10A**). Indeed, the IgM antibody response was more robust and initially increased more rapidly than IgG during this acute infection period, but the levels of antiviral IgG matched IgM by 10 dpi (**Fig 10A**). In a limiting dilution assay where we stimulated RM PBMC and screened supernatants by MAYV ELISAs, anti-MAYV antibody-secreting cells were detected with a similar frequency (~$10^1$ cells / $10^6$ PBMC) in all three animals at 10 dpi (**Fig 10B**). Furthermore, we utilized these same LDA supernatants in MAYV neutralization assays to compare the frequency of cells secreting MAYV-binding versus -neutralizing antibodies and found

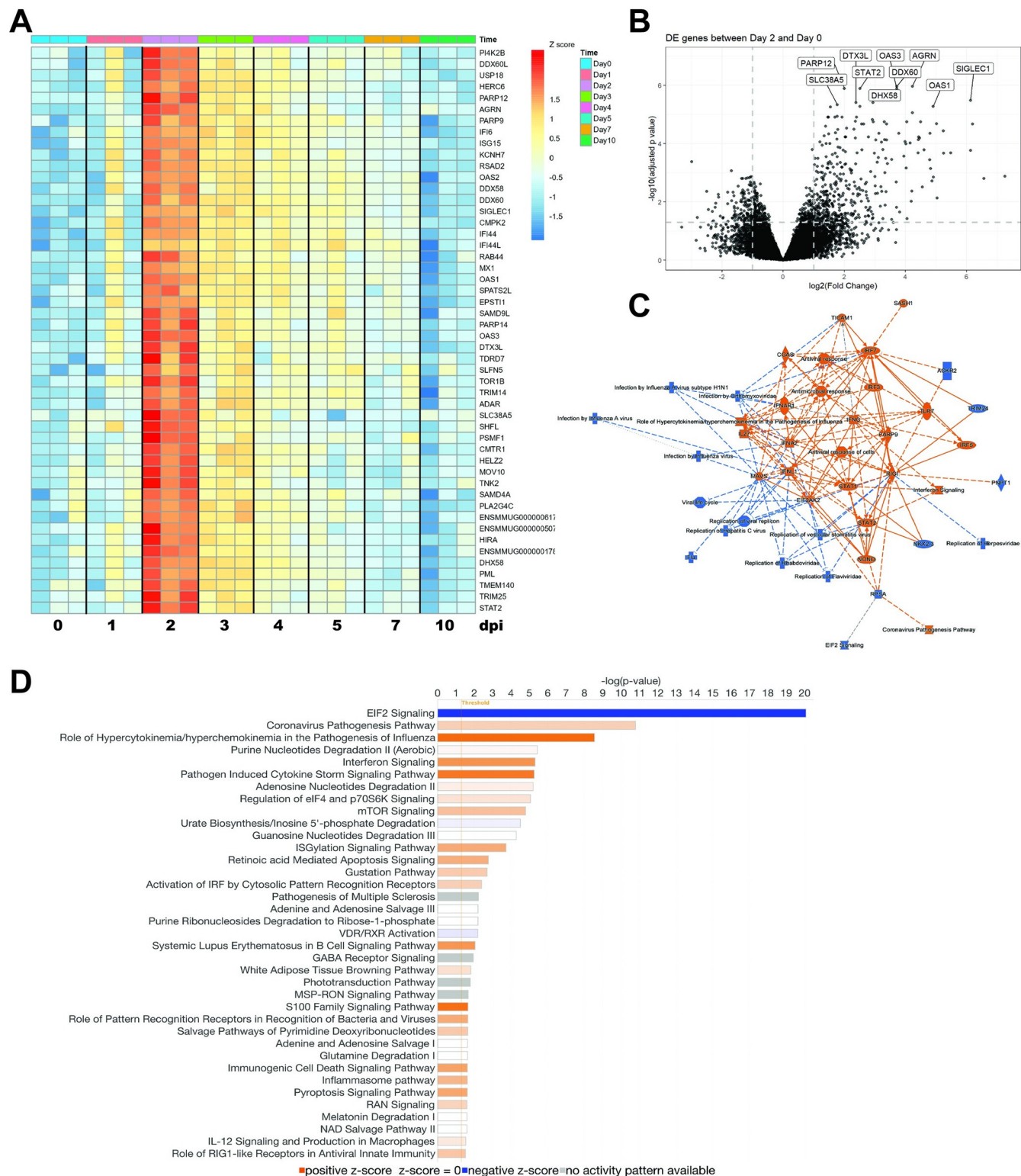

**Fig 9. Transcriptional analysis of changes following MAYV infection and pathways analysis.** (**A**) Heat map of top 50 differentially expressed (DE) genes between 0 and 2 dpi (FDRp<0.05 and |FC|>2). (B) Volcano plot of top DE genes defined in (A) between 0 and 2 dpi with the top 10 genes annotated in the plot. (C) Graphical summary of the top hits for pathways and transcripts that are altered between 0 and 2 dpi (FDRp<0.2 and |FC|>1.5) generated using Ingenuity Pathway Analysis software. (D) Pathway analysis of the top 37 enriched pathways between 0 and 2 dpi (FDRp<0.2 and |FC|>1.5). Colors in all plots encode z-scores that are more upregulated in red/orange or more downregulated in blue.

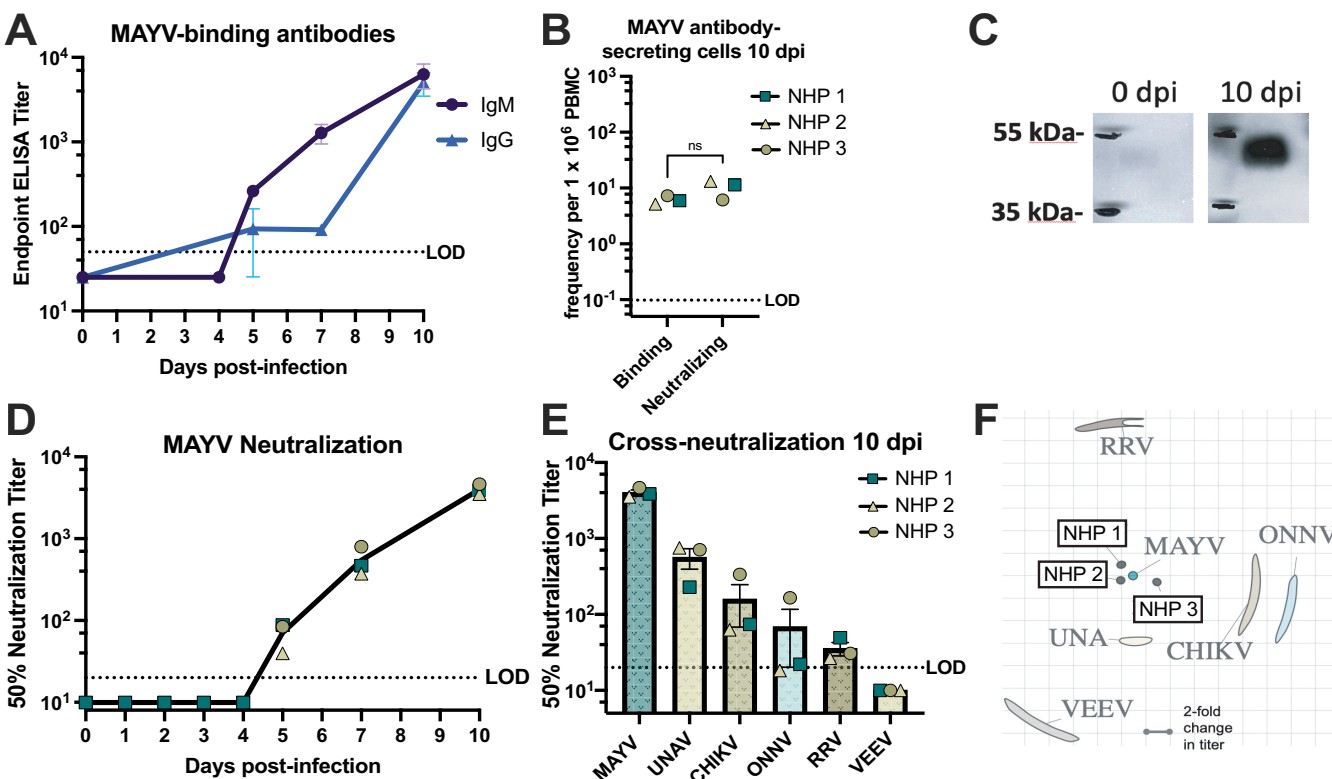

**Fig 10. Characterization of MAYV-specific antibodies and analysis of cross-reactive breadth.** (**A**) The development of MAYV-binding, IgM and IgG isotype antibody titers were quantified in macaque plasma at 0, 5, 7, and 10 dpi in ELISA. The LOD was a 1:50 plasma dilution with undetectable values graphed as half the LOD. (**B**) The frequency of cells secreting MAYV binding or neutralizing antibodies were quantified in limiting dilution assays in which macaque PBMC from 10 dpi was stimulated with IL-2 and R848 and supernatants were screened in either MAYV ELISAs or MAYV neutralization assays. The LOD frequency was 0.01 cells per $1 \times 10^6$ PBMC. (**C**) The binding specificity of MAYV-specific antibodies was characterized in a western blot in which inactivated, purified MAYV was ran on a 4–12% Bis-Tris gel and probed with macaque plasma from 0 or 10 dpi. Blots from only one animal are shown but are representative for all three animals. (**D**) The longitudinal development of MAYV-neutralizing antibodies was quantified in MAYV neutralization assays using heat-inactivated macaque plasma at 0–5, 7, and 10 dpi. 50% plaque reduction neutralization titers ($PRNT_{50}$) were determined in non-linear regression. The LOD was a 1:20 plasma dilution and undetectable values were graphed as half of the LOD. (**E**) The breadth of antibodies that neutralized other relevant alphaviruses following MAYV infection were characterized in cross-neutralization assays against UNAV, CHIKV, ONNV, RRV, and VEEV using heat-inactivated macaque plasma from 10 dpi. The LOD was a 1:20 plasma dilution and undetectable values were graphed as half of the LOD. (**F**) Antigenic cartography mapping the antigenic distances between viruses is used to visualize the cross-neutralization data. Error bars are SEM in (**A**) and (**E**). A paired t test was used to compare frequency of MBC secreting MAYV binding and neutralizing antibodies in (**B**).

that these occur at about the same frequency (6 cells / $10^6$ PBMC binding vs 10 cells / $10^6$ PBMC neutralizing; p = 0.2703) (**Fig 10B**). To interrogate the breadth of the MAYV-specific antibodies, we probed immunoblots of purified MAYV particle preparations with RM plasma from 0 and 10 dpi. Viral-envelope specific antibodies were detected in all RMs (**Fig 10C**). MAYV-neutralizing antibodies were detected as early as 5 dpi in all three macaques using plaque reduction neutralization assays (**Fig 10D**). These neutralizing antibody levels reached 50% plaque reduction neutralization titers ($PRNT_{50}$) of $3.5 \times 10^3$-$4.6 \times 10^4$ by 10 dpi (**Fig 10D**). Finally, antiviral breadth of neutralizing antibodies was determined using plaque neutralization assays against additional Semliki Forest antigenic complex viruses including UNAV, CHIKV, O'nyong'nyong virus (ONNV), and Ross River virus (RRV) as well as Venezuelan equine encephalitis virus (VEEV) (**Fig 10E**). Pre-infection plasma was screened to ensure the absence of pre-existing cross-neutralizing antibodies, which were found to be devoid of any neutralizing activity ($PRNT_{50} < 20$) against any of the viruses tested. At 10 dpi, cross-neutralizing antibodies were detected for UNA, CHIKV, ONNV, and RRV but not VEEV (**Fig 10E**).

Cross-neutralization at 10 dpi was greatest against viruses more antigenically related to MAYV, which is visualized using antigenic cartography (**Fig 10F**). RM plasma clustered around MAYV due to highest neutralization potency with UNA, CHIKV, and ONNV positioned nearer to this cluster, but RRV and VEEV positioned further away due to little or no detectable neutralization against these viruses (**Fig 10F**). In summary, our findings indicate that MAYV-specific antibodies develop as early as 5 dpi and expand in both magnitude and breadth, with the capability to neutralize other related arthritogenic alphaviruses.

## Discussion

MAYV is a virus endemic to Central and South America that is considered an emerging public health threat. While MAYV-specific therapeutics have been reported in the literature, their evaluation has been constricted to mouse models of infection due to lack of a fully defined NHP model. In this investigation, we characterized MAYV infection in rhesus macaques to better understand viral dissemination, pathogenesis and immunity. Before initiating our RM study, we compared the pathogenicity of MAYV strains and related UNAV in both immuno-competent and immunodeficient mice. While UNAV replicates more quickly in IFNαR-/- mice leading to earlier demise when compared to the MAYV strains, we found that MAYV$_{BeAr505411}$ infection resulted in the most robust viral replication in WT mice, which informed our strain selection for use in macaques. It should be noted that varying passage history of the MAYV strains used in our study may impact our conclusions regarding murine pathogenesis and strain selection. For example, MAYV$_{TRVL}$ has been extensively passaged, which may have contributed to reduced virulence in mice. Nevertheless, our data supports increased pathogenesis of the MAYV$_{BeAr505411}$ strain in mice relative to the other strains that were tested here. In 1967, MAYV-infected NHPs were reported to develop viremia lasting 4–5 days [67]. In our study, we explored the kinetics of MAYV viremia between 1 and 10 dpi, identifying the duration of viremia to be between 4 and 7 days with peak viral RNA levels occurring at 2 dpi. We isolated infectious virus from RM plasma from 1–4 dpi, suggesting that there is a brief window for blood-borne transmission. Future studies will be required to validate transmission potential and to evaluate disease presentation beyond the initial control of viremia.

In our study, we explored MAYV tissue tropism in a wide breadth of anatomical sites through quantification of viral genomes and qualification of inflammation via histopathology. Previously, the characterization of MAYV tissue tropism has been largely derived from infection in mouse models and mammalian cell lines [25,51,77,95,96]. Our study in NHPs indicated that MAYV efficiently disseminated throughout major organ systems, infecting a broad range of muscles, joints, nerves, lobes of the brain, compartments of the heart, lymphoid tissues, and other primary organs. Our evidence of viral detection and lymphocyte aggregation near rare blood vessels of the heart and central nervous tissues is in agreement with experimental CHIKV infections CHIKV and clinical outcomes in patients infected with CHIKV [39,40]. While we isolated infectious virus in multiple tissue types at 10 dpi, it is unclear whether this will lead to sustained viral replication in joints and muscles and/or be responsible for chronic disease symptoms in MAYV-infected humans. Robust MAYV viremia and widespread tissue distribution to the distal joints and muscles indicated that the virus is capable of causing disease in multiple tissues. While we did not detect overt clinical signs of arthritic or neurologic disease, there is potential for chronic disease development beyond 10 dpi in this model as evidenced by our viral detection data and pathological changes associated with infection.

A paucity of published data exists on the histopathology of MAYV in humans, presumably due to few cases causing mortality, difficulty in obtaining biopsy samples, and presence of other established diagnostics. This highlights the importance of elucidating the microscopic

changes caused by MAYV in an animal model with high anatomic similarity to humans. Appurtenant to other techniques utilized, our study is the first to explore the pathology induced by MAYV infection in a broad range of rhesus macaque tissues, which has significant implications for understanding viral pathology in humans. Arenívar and colleagues have reported MAYV arthralgia to occur commonly in the hand, knee, ankle/foot, wrist, elbow, and shoulder in decreasing frequencies, which is a significant cause of disability in humans [97]. Equivalently, in the subacute period of infection, CHIKV frequently affects the distal joints of the limbs and may involve the elbows and knees [36]. Microscopic analysis of muscle and joint biopsies from alphavirus-infected patients has been employed for diagnosis in addition to molecular techniques. In CHIKV-infected humans, some of these microscopic findings have included synovial hyperplasia, muscle degeneration and necrosis, and mononuclear to mixed inflammation with indication of a change in cellular infiltration profiles between acute, sub-acute, and chronic infections [98,99]. Of the extensive tissue sets sampled in our study, the fingers, wrists, ankles, and toes were the most consistent sites for inflammation and perivascular lymphocytic infiltration. Synovial and endothelial cell hyperplasia also occurred in these peripheral joints across the three animals. Similar findings were present in the knees of two animals and the elbows of one, which is consistent with what has been noted for CHIKV infection in humans, NHPs, and mouse models [74]. Previous studies in multiple mouse strains have extensively characterized joint and muscle tissue inflammation following MAYV infection in the footpads [25,51,77]. Mimicking the results from our study, BALB/c mice displayed inflammatory infiltrates of the ligament, tendon, and muscle surrounding joints at 10 dpi [77]. Other studies have described vasculitis, mononuclear infiltration, polymorphonuclear cell infiltration, muscular necrosis and inflammation, and dermal edema [44] throughout the course of disease [46]. Despite investigation in only three animals, the consistent pathologic findings of inflammation in the peripheral joints of these rhesus macaques, in conjunction with viral detection, enhances our understanding of pathogenesis in an appropriate animal model. Taking into account alternative study designs with respect to timing, successive studies may supplement the musculoskeletal pathology information by inclusion of a larger subset of joints and muscles, including axial structures, to determine the extent of inflammation and screen for any potential tropism between appendicular versus axial structures.

Another discovery homologous with human MAYV infection was a maculopapular rash at 10 dpi spanning the caudal ventrum of NHP 3. In humans, papular to maculopapular rash described as variably pruritic typically presents on 5 dpi and generally lasts 3–7 days following onset [34]. Recorded spread of the rash is generally on the limbs and trunk [100,101] and active replication of MAYV in human skin has been observed up to 4 days after infection [102]. In one macaque in our study, a rash was found at 10 dpi and was only able to be fully visualized with shaving, which would have precluded identification at any prior timepoints particularly given that pruritis was not a feature. The histologic picture matched the gross presentation with increased severity of lymphocytic dermatitis and epidermal hyperplasia in areas of macules and papules. Perivascular inflammation extended into distant areas of the integument on the thorax and were also found in one other animal. Maculopapular rash has not been reported in mice although it is possible it is missed without shaving the hair from these animals. Biopsies of human specimens are not widely conducted, potentially making the macaque a uniquely significant model for investigating dermatologic presentations of MAYV.

In our study, other sampled sites were recognized through the conjunction of histologic lesions and positive viral identification, which may offer promising insight into processes occurring in humans. On routine microscopic evaluation, minimal to mild perivascular leukocytic aggregates spanned multiple organ systems. Influence of any potential age-related or incidental pre-existing lesions could not be definitively elucidated utilizing H&E-stained

sections. However, in CHIKV infections in humans, it has been established that pre-existing chronic conditions are associated with increased inflammation and worsened disease [103,104]. Future investigation into potential colocalization of viral particles and inflammatory or degenerative foci, which was not within the scope of this current study, would improve identification of lesion relevance.

The liver is a tissue of interest as it, along with the spleen, is considered a primary site of viral replication and the Pan American Health Organization recommends histologic and immunohistochemical analysis of both tissues [96,105]. It was demonstrated that oxidative stress causes tissue damage in BALB/c mice, which manifests as polymorphonuclear hepatitis from 1 to 7 dpi [96]. As with other tissues, we observed a predominance of mononuclear inflammation in the liver of two rhesus macaques, with one having a pre-existing chronic hepatopathy. Though extensive determination of potential neurotropism in non-human primate species has yet to be carried out, MAYV possesses the ability to infect human neural cells with meningoencephalitis being described in rare human cases [106], and neurotropism has been demonstrated in both wild-type and immunocompromised mice [44]. We identified small leukocytic foci within different central and peripheral nervous system components between our experimental subjects, that was consistent with viral detection and these features bear additional probing as proposed for other tissues.

Coinciding with peak viremia at 2 dpi, we observed the elevation of proinflammatory cytokines and chemokines that have been associated with persistence of disease symptoms, although these responses could also play more of a protective role [78,107]. It should be noted that we detected more limited levels of proinflammatory cytokine and chemokine responses in one animal (NHP 2) although other evidence of activation of innate immunity was present. In fact, a key component of innate immune activation that we characterized was the consistent activation of monocytes (classical, non-classical, intermediate), dendritic cells (myeloid and plasmacytoid) and NK cells between 2 and 4 dpi, which returned to baseline activation status by 10 dpi, closely mirroring viremia kinetics. To identify early adaptive immune responses, we used flow cytometry to detect changes in naïve and memory T cell population frequencies and capture their cytotoxic and proliferative functions in response to MAYV infection. We identified CD4+ and CD8+ memory T cells with proliferative (Ki67+) and cytotoxic functions (granzyme B+) that expanded following infection, which makes it likely that they target MAYV-infected cells. However, we did not have access to a MAYV peptide library, but our future studies will characterize virus-specific T cell responses. Our transcriptomics data also indicates the robust activation of interferon responses coinciding with peak viremia as well as upregulation of pathways with antiviral effects, which is consistent with RNA-seq data for CHIKV infection comparing mouse and human gene expression profiles that showed similar signatures of immune activation [108–111]. Lastly, we characterized the timing of humoral immunity during acute infection, which indicated the presence of virus binding and neutralizing antibodies as early as 5 dpi, with breadth extending to similar arthritogenic alphaviruses as early as 10 dpi. We hypothesize that these cross-neutralizing antibody responses will expand in magnitude as the adaptive immune response develops beyond 10 dpi as we have observed in CHIKV-infected patients. Although antibody cross-reactivity within the SFV antigenic complex is well established, major questions remain regarding protective levels of cross-reactive antibody titers following infection and the duration of this immunity [23].

In this study, we were only able to explore MAYV pathogenesis and immunity in three macaques. With a small animal number, it is difficult to capture the spectrum of disease, although, many of our virologic, immunologic and histologic findings were consistent among all three animals. Limited tissue sampling could bias tissue viral load and histologic analyses as the whole tissue cannot be assayed in entirety, which is a limitation that should be considered

when interpreting the tissue viral load and pathology data. Sex and age-related variation are two additional variables that were not addressed in this study due to small animal number but are variables that have been found to impact CHIKV disease [66,112,113]. Future MAYV NHP studies should explore both a shorter study duration to capture acute tissue viral loads and examine tissue-resident inflammatory immune responses as well as a longer study duration to understand long term kinetics and duration of homotypic and heterotypic adaptive immunity. This study establishes an MAYV infection model in NHP that contributes to our understanding of pathogenesis and immunity that could be used for the evaluation of MAYV-specific vaccines, monoclonal antibody therapies, and antivirals.

## Materials and methods

### Ethics statement

Mice were housed in the ABSL-3 facility at the Vaccine and Gene Therapy Institute (VGTI) of Oregon Health and Science University (OHSU) in ventilated racks with open access to food and water with a 12-hour light/dark cycle. Mouse experiments were performed in compliance with the Oregon Health and Science University (OHSU) Institutional Animal Care and Use Committee (IACUC Protocol #0913). Rhesus macaque studies were performed in an ABSL-2 facility at the Oregon National Primate Research Center (ONPRC) (IACUC #0993). Both facilities are accredited by the Association for Accreditation and Assessment of Laboratory Animal Care (AALAC) International. Mouse and macaque experiments were performed in compliance with good animal practices outlined by local and national welfare bodies and all efforts were made to reduce pain, distress, and discomfort experience by the animals when possible. When possible, rhesus macaques were housed in pairs with visual and auditory contact of other animals for social interaction and enrichment. Animals were fed standard chow supplemented with food enrichment. Animals were euthanized according to the recommendations of the American Veterinary Medical Association 2013 Panel on Euthanasia.

### Cells and viruses

Vero cells (ATCC CCL-81) were propagated at 37°C and 5% $CO_2$ in Dulbecco's Modified Eagle Medium (DMEM; Thermo Scientific) containing 5% fetal calf serum (FCS; Thermo Scientific) supplemented with 1X penicillin-streptomycin-glutamine (PSG; Life Technologies). *Aedes albopictus* C6/36 cells (ATCC CRL1660) were grown at 28°C with 5% $CO_2$ in DMEM containing 5% FCS and 1X PSG. Alphaviruses $MAYV_{BeAr505411}$ (NR-49910), $MAYV_{Guyane}$ (NR-49911), $MAYV_{TRVL4675}$ (NR-49913), $MAYV_{Uruma}$ (NR-49914), $UNAV_{MAC150}$ (NR-49912), $ONNV_{UgMP30}$ (NR-51661), $RRV_{T-48}$ (NR-51457) and $VEEV_{TC-83}$ (NR-63) were obtained though BEI Resources. $MAYV_{CH}$ was generated from an infectious clone provided by Dr. Thomas Morrison (University of Colorado Denver) and $CHIKV_{181/25}$ was generated from an infectious clone as previously described [114]. Viruses were propagated in *Aedes albopictus* C6/36 cells. At 72 hours post-infection (hpi), clarified culture supernatants were pelleted through a 10% sorbitol cushion by ultracentrifugation at 82,755 x g for 70 minutes. The viral pellets were resuspended in PBS, aliquoted, and frozen at -80°C. Virus was tittered by limiting dilution plaque assays using confluent monolayers of Vero E6 cells. Infected cells were rocked continuously for 2 hours at 37°C and overlaid with CMC-DMEM supplemented with 5% FBS, 1X PSG, and 0.3% high / 0.3% low viscosity carboxymethylcellulose (CMC; Sigma). Plaque assays for MAYV, UNA, RRV and VEEV were fixed with 3.7% formaldehyde and stained with 0.2% methylene blue at 48 hpi; and the plaque assays for CHIKV and ONNV were fixed and stained at 72 hpi. Plaques were enumerated under a light microscope and titers of viral stocks were determined. Virus stocks used for all lab experiments were either passage 1 or 2, although

passage history at BEI prior to arrival in the lab does vary by strain and has been presented previously in a table for the MAYV strains by Powers 2006 *et al* [115].

## Mouse experiments

C57BL/6 mice were purchased from Jackson Laboratories and interferon alpha receptor knockout (IFNαR-/-) mice originated from the OHSU/VGTI established breeding colony. MAYV and UNAV infections were performed in 4-week-old female C57BL/6 mice (n = 5 per virus group) and 13-week-old male and female IFNαR-/- mice (n = 4 per virus group). Mice were inoculated subcutaneously in the right footpad with 20μL containing $10^4$ plaque forming units (PFU) of $MAYV_{BeAr505411}$ (NR-49910), $MAYV_{CH}$, $MAYV_{Guyane}$ (NR-49911), $MAYV_{TRVL4675}$ (NR-49913), $MAYV_{Uruma}$ (NR-49914), or $UNAV_{MAC150}$ (NR-49912). Infected C57BL/6 mice were bled at 2 days post-infection (dpi) to quantify the level of viremia in serum collected from clotted blood samples. These mice were euthanized by isoflurane overdose at 5 dpi to assess viral burden in ankle, calf, quad, spleen, brain, and heart tissues. IFNαR-/- mice were bled at 1 dpi to quantify the levels of serum viremia; body weight, survival, and ipsilateral footpad swelling measurements were recorded daily. IFNαR-/- mice were euthanized when 20% of body weight had been lost.

## Nonhuman primate experiments

Three adult male rhesus macaques (*Macaca mulatta*) ages 4, 10 and 13 years were included in this study. Animals were sedated prior to any procedure. Lymphoid organ biopsies (axillary and mesenteric lymph nodes, and spleen) and blood were surgically collected at 28 days prior to infection [116–118]. Animals were infected with $10^5$ plaque forming units (PFU) of MAYV diluted in 1mL of PBS through 100μL subcutaneous injections in both of the arms and hands in an attempt to mimic virus inoculation through the bite of an infected mosquito. Animals were fed standard monkey chow with routine food supplements for enrichment. The animals were monitored daily for clinical signs of disease and discomfort. Temperature and body weight were measured on the days on which peripheral blood and urine samples were collected (0, 1–5, 7, and 10 dpi). Blood was collected for monitoring by both complete blood count and serum chemistry analyses and analytes were compared to standard reference ranges [119]. Whole blood was layered over lymphocyte separation medium (Corning) and centrifuged for 30 minutes at 2,000 rpm for plasma and peripheral blood mononuclear cell (PBMC) isolation. PBMC were washed in RPMI medium (Fisher) supplemented with 5% FBS and 1X PSG. Rhesus macaques were humanely euthanized at 10 dpi and complete necropsies were performed. Representative tissue sections (~1cm$^3$) from joint, muscle, lymphoid, major organs, nervous system, and reproductive tissue were collected into 1mL TRIzol reagent (Invitrogen) for RNA isolation or fixed in 10% formalin for histopathology. When appropriate, right and left tissues (i.e., fingers, toes, quadriceps, triceps, etc.) were combined for RNA analysis. An additional section from each tissue was preserved in RNAlater.

## Histopathological analysis

A wide range of tissues were collected at necropsy for histologic analysis, which underwent fixation in 10% neutral buffered formalin for 24 hours and then 80% ethanol, stored at 4˚C, for 24–72 hours followed by routine processing, sectioning at 5 μm, and staining with hematoxylin and eosin (HE). Slides were assessed on Leica DFV495 light microscopes by two board-certified veterinary pathologists and were scanned with a Leica Aperio AT2 slide scanner for creation of digital images. Presence and relative intensity of lymphocytic inflammation was graded based on a scale of—to +++ (**Tables 2 and S1**) within all non-hematopoietic tissues. Lower

scores (+) indicated one small aggregate of perivascular lymphocytes and ranged up to inflammation affecting the majority of blood vessels, in small to moderate numbers, with or without infiltration into the surrounding tissue (**Tables 2 and S1**). Any additional pathologic diagnoses were included in these tables as well as separately for the hematopoietic tissues (**S2 Table**).

## Viral RNA detection

Mouse tissues were homogenized in 1mL of 1X PBS with approximately 250μL of silica beads (VWR 48300–437) using a bead beater for three cycles of 45 seconds on and 30 seconds off (Precellys 24 homogenizer, Bertin Technologies). Samples were centrifuged at 5,000 rpm for 5 minutes in a microfuge to remove cellular debris, and 300μL of each homogenate was removed for RNA isolation. Nucleic acids from mouse tissues were isolated using the Promega Maxwell 48 sample RSC automated purification system and the Maxwell RSC Viral TNA extraction kit (Promega). Total nucleic acids were resuspended in 60μL of RNAse free water. RM tissue samples were homogenized in 1mL of TRIzol reagent (Invitrogen) with approximately 250μL of silica beads using a Precellys 24 homogenizer bead beater as described above. Samples were centrifuged at 5,000 rpm for 5 minutes to remove cellular debris. Total RNA was isolated from either 200μL of homogenized tissue or 200μL of plasma or urine using a Direct-zol RNA Miniprep Plus kit (Zymo Research) following the manufacturer instructions. Total RNA was resuspended in 50μL of RNAse-free water. Prepared RNA was quantified using a Nanodrop and diluted to 100ng/μL. Contaminating DNA was removed from all of the RNA samples by digestion with ezDNase (ThermoFisher). Single stranded cDNA was generated from 1μg of total RNA using random hexamers and reverse transcriptase Superscript IV (Invitrogen) following the manufacturer's protocol. Gene amplicons served as quantification standards. The following primers and probe were used to detect MAYV RNA: Forward- CCATGCCGTAACGATT GC, Reverse- CTTCCAGGCTGCCCGGCACCAT, and probe FAM- TGGACACCGTTCGA-TAC–MGB. The following primers and probe were used to detect UNAV RNA: Forward-GAAGCTTTTGTCTCCGGTGAA, Reverse-ATGACAATGGCCCGAATATGA, and Probe-FAM-TGAATGTCGCTGGGACT–MGB. Quantitative RT-PCR was performed on a Quant-Studio 7 Flex Real-Time PCR system. All data was analyzed using Applied Biosystems Quant-Studio 6 and 7 Flex Real-time PCR System software. For mouse tissues, viral RNA levels were normalized to a murine housekeeping gene, ribosomal protein RPS17. Viral RNA levels in RM tissues and blood were reported per μg of input RNA. All qRT-PCR reactions were performed in triplicate.

## Quantification and isolation of infectious virus

Limiting dilution plaque assays were used to quantify viral loads in tissues and blood. For this assay, aliquots of 20μL of tissue homogenate, tissue culture supernatant, or mouse serum were serially diluted 10-fold in DMEM containing 5% FBS and 1X PSG, which was added to confluent monolayers of Vero cells in 48-well plates. The plates were rocked for 2 hours at 37˚C and then CMC-DMEM was added to each well. At 2 dpi, the plates were fixed with 3.7% formaldehyde and stained with 0.2% methylene blue for microscopic visualization and enumeration of the plaques.

Isolation of MAYV from mouse tissues was carried out as previously described [25]. MAYV was isolated from NHP plasma and tissues as previously described for CHIKV [74]. Briefly, tissues were collected in 1mL of 1X PBS containing approximately 250μL of silica beads (VWR 48300–437) and homogenized using a bead beater for three cycles of 45 seconds on and 30 seconds off (Precellys 24 homogenizer, Bertin Technologies). Samples were centrifuged at 5,000 rpm for 5 minutes to remove cellular debris, sterile-filtered (0.22μM filter), and 400μL was used to infect a T25 flask of confluent C6/36 cells. At 3 dpi, supernatants were collected from C6/36 cultures and

tittered in triplicate by limiting dilution plaque assays on Vero E6 cells as described above. Samples were considered positive for infectious virus if one or more plaques were detected, providing a limit of detection of 3.3 PFU/mL of cellular supernatant.

### Transcriptomic analysis

Total RNA from rhesus macaque PBMC isolated using the TRIzol extraction method described above was prepared for transcriptomic analysis using the Illumina TruSeq Stranded mRNA Library Prep Kit (RS-122-2101, Illumina) as previously described [120]. The library was validated using an Agilent DNA 1000 kit on a bioanalyzer. Samples were sequenced by the OHSU Massively Parallel Sequencing Shared Resource using an Illumina NovaSeq.

Differential expression analysis was performed by the ONPRC Bioinformatics & Biostatistics Core. The quality of the raw sequencing files was evaluated using FastQC [121] combined with MultiQC [122] (http://multiqc.info/). Trimmomatic [123] was used to remove any remaining Illumina adapters. Reads were aligned to Ensembl's Mmul_10 genome along with its corresponding annotation, release 109. The program STAR [124] (v2.7.10b_alpha_220111) was used to align the reads to the genome. STAR has been shown to perform well compared to other RNA-seq aligners [125]. Since STAR utilizes the gene annotation file, it also calculated the number of reads aligned to each gene. RNA-SeQC [126] and another round of MultiQC were utilized to ensure alignments were of sufficient quality.

Gene-level raw counts were filtered to remove genes with extremely low counts in many samples following the published guidelines [127], normalized using the trimmed mean of M-values method (TMM) [128], and transformed to log-counts per million with associated observational precision weights using the voom method [129]. Gene-wise linear models with primary variable day after infection, and accounting for within subject correlation, were employed for differential expression analyses using limma with empirical Bayes moderation [130] and false discovery rate (FDR) adjustment [131]. Differential expression data were analyzed through the use of IPA (QIAGEN Inc., https://www.qiagenbioinformatics.com/products/ingenuity- pathway-analysis), using a stringent cutoff for significant molecules of FDRp $< 0.2$ and $|FC| > 1.5$. The background reference set used was the dataset of all genes in the differential analysis.

### Neutralization assays

RM plasma was heat inactivated for 30 minutes at 56˚C and serially diluted in DMEM supplemented with 5% FBS and 1X PSG. Diluted plasma was mixed with media containing approximately 70–120 plaque forming units of $MAYV_{BeAr505411}$, $CHIKV_{181/25}$, $UNAV_{Mac150}$, $ONNV_{UgMP30}$, $RRV_{T-48}$, or $VEEV_{TC-83}$. Samples containing plasma and virus were incubated for 2 hours at 37˚C with 5% $CO_2$ with continuous rocking and then transferred to 12-well plates of confluent Vero cells. Plates were incubated for an additional 2 hours at 37˚C with continuous rocking followed by addition of a CMC-DMEM overlay. Plates were incubated 48 hours for MAYV, UNAV, RRV and VEEV or 72 hours for CHIKV and ONNV, then cells were fixed and stained as described above. The 50% plaque neutralization titers ($PRNT_{50}$) were calculated by non-linear regression analysis using GraphPad Prism 9 software after determining the percent of plaques at each dilution relative to control wells containing no plasma.

### Antigenic cartography

The antigenic cartography plot to visualize alphavirus cross-neutralization following MAYV NHP infection was assembled as previously described [132,133] and implemented using the Acmacs Web Cherry platform (https://acmacs-web.antigenic-cartography.org/). To ultimately

construct the antigenic map, a table of calculated antigenic distances ($D_{ij}$) between each viral antigen ($i$) and plasma sample ($j$) using plasma titers for each plasma-titer pair ($N_{ij}$) is generated. To calculate table distance, the titer against the best neutralized virus for that plasma sample is defined as $b_i$ and the distances from each virus for that plasma are calculated as $D_{ij} = log_2(b_i)\text{-}log(N_{ij})$. For the highest neutralization titer for a plasma sample, $N_{ij} = b_i$, and the distance will be equal to 0. For the remaining plasma-virus pairs, table distance $D_{ij}$ is equivalent to the fold-difference in titer between $b_{ij}$ and $N_{ij}$. Euclidean map distance ($d_{ij}$) for each plasma-virus pair is found by minimizing the error between the table distance $D_{ij}$ and map distance, $d_{ij}$, using the error function $E = \Sigma_{ij}e(D_{ij},d_{ij})$, where $e(D_{ij},d_{ij}) = (D_{ij}\text{-}d_{ij})^2$ when the neutralization titer is detectable or above 1:20. For instances where no detectable plasma neutralization titer is observed for a virus with neutralization titers <1:20, values of 19 are entered and the error is defined as $e(D_{ij},d_{ij}) = (D_{ij}\text{-}1\text{-}d_{ij})^2(1/1+e^{-10(Dij-1-dij)})$. To make a map and derive $d_{ij}$ for each plasma-virus pair, viruses and plasma samples are assigned random starting coordinates and the error function is minimized using the conjugate gradient optimization method. Each square grid line on the antigenic map represents a two-fold change in plasma neutralization titer.

## Enzyme-linked immunoassays (ELISA)

Purified MAYV$_{BeAr505411}$ was inactivated at 56˚C for 30 minutes, diluted in 1X PBS, and $5\times10^8$ plaque forming units (PFU) were added to each well of 96-well high binding plates (Corning) and incubated for 4 days at 4˚C. To detect total IgG by ELISA for limiting dilution assays described blow, a goat anti-human IgG (H+L) coating antibody (Jackson Immuno Research) was diluted in 1X PBS and added to the 96-well high binding plates at 1 µg/mL. Plates were washed with ELISA wash buffer (0.05% Tween-20, 1X PBS) and blocked for 1 hour with ELISA wash buffer containing 5% milk. The plates were washed with ELISA buffer and then 100µL of 1:3 serial dilutions of heat-inactivated RM plasma were added and incubated for 1 hour. Plates were washed with ELISA wash buffer before secondary anti-monkey IgG or IgM (H+L) HRP-conjugated detection antibodies (Rockland) were diluted 1:5,000 and added to appropriate plates. Plates were washed, developed with OPD substrate buffer (0.05M citrate, 0.4 mg/mL o-phenylenediamine, 0.01% hydrogen peroxide, pH 5), and reactions were stopped with 1M HCl. A BioTek plate reader was used to read plates at 490nm. Log-log transformation of the linear portion of the curve was performed and 0.1 OD units was the cut-off point to calculate end point titers.

## Limiting dilution assay quantification of MAYV antibody-secreting cell frequency

Limiting dilution assays (LDA) to characterize the frequency of antibody-secreting cells, previously defined as memory B cells, were carried out as previously described [134]. Briefly, RM PBMC collected at 10 days post-infection (dpi) were resuspended in Roswell Park Memorial Institute (RPMI) 1640 medium supplemented with 5% FBS and 1X PSG. We chose not to refer to cells in our assay at 10 dpi as memory B cells because at this time following infection, this may also include more premature plasmablasts. Two-fold serial dilutions of PBMC were added to a 96-well round-bottom plate; the top row contained $3\text{--}5\times10^5$ PBMC per well. Next, 100µL of RPMI stimulation media containing 5% FBS, 1X PSG, 2.5 µg/mL R848 (InvivoGen), and 1000 U/mL IL-2 (Prospec) was added to each well with the exception of an unstimulated control column containing PBMC only. The 96-well plates were incubated for 7 days at 37˚C with 5% $CO_2$, and then culture supernatants were collected for analysis by IgG ELISA detecting either total IgG (to determine the frequency of antibody producing cells) or MAYV

proteins (to determine the frequency of viral antigen specific antibody producing cells) [135]. The supernatants from unstimulated PBMC served to normalize against background absorbance values. LDA supernatants were also collected for quantification of cells secreting MAYV-neutralizing antibodies. For these assays, remaining LDA supernatants were used in MAYV$_{BeAr505411}$ neutralization assays as described above with approximately 80μL of supernatant serving in place of plasma. Neutralization in each individual well was calculated relative to a well containing MAYV only, with no LDA supernatant. Wells exhibiting 50% or greater neutralization relative to the control well were determined to be positive for neutralizing activity. The percentage of negative wells (below 50% neutralization) vs cell count in each row was graphed to calculate the frequency of cells secreting MAYV-neutralizing antibodies.

## Plasma cytokine and chemokine analysis

The macaque inflammatory cytokine profile was characterized using a Cytokine Monkey Magnetic 29-plex Panel for Luminex Platform Kit (Invitrogen) according to the manufacturer's instructions using a 7-point standard curve. First, 25μL of RM plasma was incubated for 2 hours with beads and then washed and labeled with a biotinylated antibody for 1 hour. Beads were washed and incubated with R-Phycoerythrin conjugated to streptavidin for 30 minutes, then washed for a final time. Inflammatory cytokine levels were then quantified using a Luminex 200 Detection system (Luminex).

## Lymphocyte phenotypic analysis

RM lymphocytes isolated from peripheral blood, spleen and lymph nodes and spleen were thawed and resuspended in RPMI medium supplemented with 10% FBS and 1X PSG. Cells were pelleted by centrifugation (2,000 rpm) and washed with 1X PBS and approximately one million cells were aliquoted for each of three panels for phenotypic analysis by flow cytometry. For T cell analysis, cells were stained for cellular differentiation markers CD3, CD4, CD8, CD25, CD28, CD95, CD127, and intracellular Ki67 using fluorophore-conjugated antibodies. Naïve CD4+ or CD8+ T cells were defined as CD28+/CD95-, central memory CD4+ or CD8 + T cells were defined as CD28+/CD95+, and effector memory CD4+ or CD8+ T cells were defined as CD28-/CD95+. For B cell analysis, cells were stained with fluorophore-conjugated antibodies directed against CD3, CD20, CD27, CD14, IgD, and intracellular Ki67. Naïve B cells were defined as IgD+/CD27-, MZ-like B cells were defined as IgD+/CD27+, and memory B cells were defined as IgD-/CD27+. For innate immune cell analysis, cells were stained with CD3, CD8, CD14, CD16, CD11c, HLA-DR, CD56, CD123, and CD169 used as a marker for cellular activation. Monocytes and macrophages were defined as CD3-/CD20-/CD8-/HLA-DR + with classical monocytes being CD16-/CD14+, intermediate monocytes being CD16+/CD14 +, and non-classical monocytes being CD16+/CD14-. Dendritic cells (DCs) were defined as CD3-/CD20-/CD8-/HLA-DR+/CD16-/CD14- with myeloid DCs being CD11c+/CD123- and plasmacytoid DCs being CD11c-/CD123+. Sample analysis was performed using an LSRII instrument (BD Pharminogen) and analyzed with FlowJo Version 10 software.

## Western blot analysis

Purified MAYV$_{BeAr505411}$ proteins were separated by SDS-PAGE using 4–12% Bis-Tris polyacrylamide gels (Invitrogen) and loading ($5\times10^9$ plaque forming units/lane). Proteins were transferred to an activated PVDF membrane (Millipore) using a semi-dry transfer system (30 minutes at 25V). Membranes were blocked with 3% BSA/TBST for 1 hour and probed with a 1:250 dilution of primary RM plasma from 0 or 10 dpi. Membranes were washed with TBST and probed with a secondary IgG anti-monkey, HRP conjugated antibody (Rockland) diluted

1:10,000. Membranes were washed a final time and developed in a Pico luminescence developer solution (ThermoFisher) and exposed on X-ray film.

## Statistical analysis

Statistics and graphs were created with GraphPad Prism 9. A one-way ANOVA was used to compare means of viral RNA and viral titers levels between groups of mice infected with the different strains of MAYV. Neutralizing antibody titers were calculated using normalized variable slope non-linear regression with upper and lower limits of 100 and 0, respectively. Paired t tests were used to compare cell phenotype changes and cytokine levels at various timepoints to baseline (0 dpi).

## Supporting information

**S1 Table. Perivascular lymphocytic inflammation in endocrine, respiratory, alimentary, hepatobiliary and pancreatic, and genitourinary tissues in MAYV-infected rhesus macaques at 10 dpi.** Presence and relative intensity (+ to +++) of lymphocytic inflammation within tissues were assessed using the following scale:—, absence of pathology within sections; +, one small aggregate of perivascular lymphocytes; ++, multiple blood vessels within one or two areas of the tissue with small to moderate numbers of perivascular lymphocytes; +++, perivascular lymphocytes affecting a majority of blood vessels in small to moderate numbers with or without infiltration of the surrounding tissue. *, Chronic hepatic degeneration and regeneration; **, Rare attenuated cortical tubules, scant cellular or proteinaceous casts with few associated lymphocytes; and ***, diffuse chronic mild lymphocytic and neutrophilic urethritis.
(DOCX)

**S2 Table. Hematopoietic pathology in MAYV-infected rhesus macaques at 10 dpi.** Table summarizes pathologic diagnoses in given lymphoid tissues. Absence of observed pathology within the tissue is denoted (-).
(DOCX)

**S1 Data. Raw transcriptomics data.** Differential expression raw data for transcriptomic analysis of PBMC samples
(XLSX)

**S2 Data. Master raw data file for the manuscript.** Raw data is provided for both main and supplemental figures throughout the paper.
(XLSX)

**S1 Fig. MAYV strains in IFN$\alpha$R-/- mice.** Four 13-week-old IFN$\alpha$R-/- mice per group received a subcutaneous right footpad injection of $10^4$ PFU of MAYV$_{BeAr505411}$, MAYV$_{CH}$, MAYV$_{Guyane}$, MAYV$_{TRVL}$, MAYV$_{Uruma}$, or UNAV$_{MAC150}$. Mice were bled at 1 dpi for peak serum viremia and body weights and footpad swelling were recorded daily until animals were euthanized due to excessive loss of body weight. Serum titer of infectious virus measured by plaque-forming units per mL (PFU/mL) is log-transformed and shown in (**A**). Statistical analysis for comparison of viral titers was completed using a one-way ANOVA with log transformed data, where **** $p < 0.0001$, *** $p = 0.0001$, ** $p < 0.001$, * $p < 0.05$, ns $p > 0.05$. The Kaplan-Meier survival curve is shown in (**B**) for the four-day monitoring period until mice succumbed to infection. A Kruskal-Wallis test was used to compare survival data for the groups of mice and the only significant comparison was survival of MAYV$_{TRVL}$-infected compared to UNAV$_{MAC150}$ -infected mice, $p < 0.0001$. Footpad swelling (mm) in the ipsilateral footpad is shown in (**C**) and percent change from starting weight (%) is shown in (**D**). Error

bars are SEM when included.
(EPS)

**S2 Fig. Complete Blood Count (CBC) data for macaques over the duration of the study.**
CBC analytes from EDTA-treated whole blood: white blood cell count (**A**), lymphocytes (**B**),
neutrophils (**C**), monocytes (**D**), eosinophils (**E**), basophils (**F**), red blood cells (**G**), hematocrit
(**H**), hemoglobin (**I**), mean corpuscular volume (**J**), mean corpuscular hemoglobin (**K**), platelets (**L**), and mean platelet volume (**M**).
(EPS)

**S3 Fig. Serum chemistry panel analytes for macaques during the study.** Analytes for serum
chemistry at all blood draw timepoints included total protein (TP), albumin (ALB), alkaline
phosphatase (ALKP), alanine transaminase (ALT), aspartate transaminase (AST), gamma-glutamyl transferase (GGT), total bilirubin (TBIL), glucose (GLU), blood urea nitrogen (BUN),
creatinine (CREA), potassium (K), sodium (NA), chloride (CL), magnesium (MG), phosphorus (PHOS), cholesterol (CHOL), and triglyceride (TRIG).
(EPS)

**S4 Fig. Lymphoid pathology of MAYV-infected rhesus macaques.** Macaque lymphoid tissues were collected during necropsy, fixed, paraffin embedded, sectioned, and stained with
hematoxylin and eosin (HE). Histology was examined, and select representative images are
shown for the three animals. (**A**; Bar = 1 cm) The axillary skin of NHP 2 is discolored red-tan.
(**B**; Bar = 1 cm) The axillary lymph nodes in all three animals were enlarged and erythematous.
(**C**; Bars = 500 μm, inset 50 μm) The axillary lymph nodes have mild lymphofollicular hyperplasia and medullary sinus histiocytosis with hemophagocytosis. (**D**; Bars = 500 μm, inset
300 μm) Perifollicular sinusoids are congested. (**E**; Bar = 50 μm) Perifollicular sinusoids (black
and white arrows) have reticuloendothelial hypertrophy and are engorged with macrophages,
lymphocytes, and erythrocytes. There is rare erythrophagocytosis. (**F**; Bar = 50 μm) An
increased number of neutrophils are within the red pulp (arrowheads).
(EPS)

**S5 Fig. Lymphocytic inflammation in the nervous system of a MAYV-infected rhesus
macaque.** At 10 dpi with MAYV, macaque hematopoietic tissues were collected, fixed, paraffin
embedded, sectioned, and stained with hematoxylin and eosin (HE). Extensive histology was
examined and select representative images are shown from NHP 3. (**A**; Bar = 5 mm, inset
200 μm) Minor perivascular lymphocytic inflammation within the gray-white matter junction
of the putamen, (**B**; Bar = 100 μm) the brainstem, (**C**; Bar = 100 μm, inset 50 μm) the ventral
horn of the lumbar spinal cord, (**D**; Bar = 100 μm) and the brachial plexus.
(EPS)

**S6 Fig. Cytokine and chemokine profile following MAYV infection.** Additional inflammatory cytokines and chemokines quantified in longitudinal macaque plasma that were included
in the 29-plex Luminex panel. Select cytokines and chemokines are quantified in pg/mL at
0–5, 7, and 10 dpi. Paired t tests were used for statistical analysis where baseline (d0) was compared to each of the other timepoints but did not yield any statistically significant results
($p > 0.05$).
(EPS)

**S7 Fig. Flow cytometry gating strategy for monocyte/DC/NK panel.** Gating strategy for
monocyte/DC/NK panel is shown. Monocytes and macrophages were defined as CD3-/
CD20-/CD8-/HLA-DR+ with classical monocytes being CD16-/CD14+, intermediate monocytes being CD16+/CD14+, and non-classical monocytes being CD16+/CD14-. DCs were

defined as CD3-/CD20-/CD8-/HLA-DR+/CD16-/CD14- with myeloid DCs being CD11c
+/CD123- and plasmacytoid DCs being CD11c-/CD123+. Activated cells within each subset
were defined as CD169+.
(EPS)

**S8 Fig. Flow cytometry gating strategy for T and B cell panels.** Gating strategy for T and B
cell panels are shown. Naïve CD4+ or CD8+ T cells were defined as CD28+/CD95-, central
memory CD4+ or CD8+ T cells were defined as CD28+/CD95+, and effector memory CD4
+ or CD8+ T cells were defined as CD28-/CD95+. Naïve B cells were defined as IgD+/CD27-,
MZ-like B cells were defined as IgD+/CD27+, and memory B cells were defined as IgD-/CD27
+. Proliferating (Ki67+) T and B cells and granzyme B expressing (granzyme B+) T cells within
each subset were also quantified using these gating schemes.
(EPS)

**S9 Fig. B cell phenotype and proliferation in longitudinal peripheral blood and lymphoid
tissues following MAYV infection.** Macaque PBMC from -28, 0–5, 7, and 10 dpi (**A-B**) and
lymphocytes isolated from three lymphoid tissues either one month prior to infection or 10
dpi (**C-E**) were analyzed for B cell phenotype using flow cytometry. Changes in the total longi-
tudinal frequency of naïve, memory, and MZ-like B cell subsets (**A**) as well as proliferation
within these subsets (**B**) are quantified over time. B cell proliferation of these same subsets in
the axillary LN (**C**), mesenteric LN (**D**), and spleen (**E**) is also compared at one month prior to
and 10 dpi. Lines represent mean frequencies of the three animals and error bars represent the
standard error of the mean. Longitudinal changes in total or proliferating (Ki67+) B cell sub-
sets (**A-B**) relative to baseline (0 dpi) were compared to 7 or 10 dpi using paired t tests and
yielded only p values > 0.05, ns, for naïve, marginal zone (MZ)-like, and memory B cell sub-
sets. Statistical analyses for comparison of baseline to 10 dpi cell frequencies in the lymphoid
tissues (**C-E**) were completed using two-tailed paired t tests; only significant comparisons are
shown, all other comparisons yielded ns p values > 0.05.
(EPS)

**S10 Fig.** (**A**) Heat map of top 50 DE genes between 0 and 3 dpi (FDRp<0.05 and |FC|>2). (B)
Volcano plot of top DE genes defined in (A) between 0 and 3 dpi with the top 10 genes anno-
tated in the plot. (C) Graphical summary of the top hits for pathways and transcripts that are
altered between 0 and 3 dpi (FDRp<0.2 and |FC|>1.5) generated using Ingenuity Pathway
Analysis software. (D) Pathway analysis of the top 37 enriched pathways between 0 and 3 dpi
(FDRp<0.2 and |FC|>1.5). Colors in all plots encode z-scores that are more upregulated in
red/orange or more downregulated in blue.
(EPS)

## Acknowledgments

The following reagents were obtained through BEI Resources, NIAID, NIH, as part of the
WRCEVA program: MAYV$_{BeAr505411}$ (NR-49910), MAYV$_{Guyane}$ (NR-49911), MAYV$_{TRVL4675}$
(NR-49913), MAYV Uruma (NR-49914), UNAV$_{MAC150}$ (NR-49912), ONNV$_{UgMP30}$ (NR-
51661), RRV$_{T-48}$ (NR-51457) and VEEV$_{TC-83}$ (NR-63). The authors thank Drs. Scott Hansen,
David Morrow, Andrew Sylwester, and Eric McDonald for assistance with flow cytometry.
The authors acknowledge the Integrated Pathology Core at the Oregon National Primate
Research Center (ONPRC), which is supported by NIH Awards P51 OD 011092 and
1S10OD025002-01, for preparation and scanning of histologic slides. For contribution to the
transcriptomics and RNAseq analysis, the authors acknowledge the support of the OHSU

Massively Parallel Sequencing Shared Resource (MPSSR) as well as the ONPRC Bioinformatics & Biostatistics Core, which is funded in part by NIH grant OD P51 OD011092.

## Author Contributions

**Conceptualization:** Whitney C. Weber, Caralyn S. Labriola, Nicole N. Haese, Michael K. Axthelm, Jeremy V. Smedley, Daniel N. Streblow.

**Data curation:** Whitney C. Weber, Caralyn S. Labriola, Karina Ray, Samuel Medica, Shauna Rakshe.

**Formal analysis:** Whitney C. Weber, Caralyn S. Labriola, Craig N. Kreklywich, Samuel Medica, Shauna Rakshe, Daniel N. Streblow.

**Funding acquisition:** Daniel N. Streblow.

**Investigation:** Whitney C. Weber, Caralyn S. Labriola, Craig N. Kreklywich, Karina Ray, Nicole N. Haese, Takeshi F. Andoh, Michael Denton, Samuel Medica, Magdalene M. Streblow, Patricia P. Smith, Nobuyo Mizuno, Nina Frias, Miranda B. Fisher, Aaron M. Barber-Axthelm, Kimberly Chun, Samantha Uttke, Danika Whitcomb, Michael K. Axthelm, Jeremy V. Smedley, Daniel N. Streblow.

**Methodology:** Whitney C. Weber, Caralyn S. Labriola, Craig N. Kreklywich, Karina Ray, Nicole N. Haese, Takeshi F. Andoh, Miranda B. Fisher, Suzanne S. Fei, Daniel N. Streblow.

**Project administration:** Daniel N. Streblow.

**Resources:** Victor DeFilippis.

**Supervision:** Daniel N. Streblow.

**Validation:** Whitney C. Weber.

**Writing – original draft:** Whitney C. Weber, Caralyn S. Labriola, Craig N. Kreklywich, Daniel N. Streblow.

**Writing – review & editing:** Whitney C. Weber, Caralyn S. Labriola, Karina Ray, Michael Denton, Shauna Rakshe, Suzanne S. Fei, Daniel N. Streblow.

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
