## [Decision Letter · Decision Letter 0]

22 Aug 2023

Dear Dr. Streblow,

Thank you very much for submitting your manuscript "Mayaro virus pathogenesis and immunity in rhesus macaques" for consideration at PLOS Neglected Tropical Diseases. As with all papers reviewed by the journal, your manuscript was reviewed by members of the editorial board and by several independent reviewers. In light of the reviews (below this email), we would like to invite the resubmission of a significantly-revised version that takes into account the reviewers' comments. 

We cannot make any decision about publication until we have seen the revised manuscript and your response to the reviewers' comments. Your revised manuscript is also likely to be sent to reviewers for further evaluation.

Sincerely,

Gregory Gromowski

Academic Editor

Abdallah Samy

Section Editor

Reviewer's Responses to Questions

**Key Review Criteria Required for Acceptance?**

**Methods**

-Are the objectives of the study clearly articulated with a clear testable hypothesis stated?

-Is the study design appropriate to address the stated objectives?

-Is the population clearly described and appropriate for the hypothesis being tested?

-Is the sample size sufficient to ensure adequate power to address the hypothesis being tested?

-Were correct statistical analysis used to support conclusions?

-Are there concerns about ethical or regulatory requirements being met?

Reviewer #1: The methods section is well described, with appropriate assays used to monitor the various viral and immune parameters.

Reviewer #2: Please see Summary and General Comments

Reviewer #3: (No Response)

**Results**

-Does the analysis presented match the analysis plan?

-Are the results clearly and completely presented?

-Are the figures (Tables, Images) of sufficient quality for clarity?

Reviewer #1: The results are well described with proper analysis.

Please note that Figure 10 in the pdf file is distorted (overlap of figures) and needs to be corrected.

Reviewer #2: Please see Summary and General Comments

Reviewer #3: (No Response)

**Conclusions**

-Are the conclusions supported by the data presented?

-Are the limitations of analysis clearly described?

-Do the authors discuss how these data can be helpful to advance our understanding of the topic under study?

-Is public health relevance addressed?

Reviewer #1: The conclusions are supported by the data. The limitations of the study, as well as the significance of the findings, are well discussed.

Reviewer #2: Please see Summary and General Comments

Reviewer #3: (No Response)

**Editorial and Data Presentation Modifications?**

Reviewer #1: I have a number of minor suggestions/editorial edits:

- Line 24: add (NHP) after non-human primates as this is the first time the abbreviation would be used.

- Line 28: You can mention here the animal number of n=3.

- Lines 72-73: references 12-15 look to be studies in the laboratory (instead of in the real world), so it would be good to check this and mention it accordingly.

- Line 158: I assume that instead of immunocompetent, this should be immunodeficient?

- Line 164. In the section on MAVY replication kinetics, it would be good to also mention if any animals displayed any overt signs of disease or discomfort (e.g. joint pain). As the authors do not mention it, I assume nothing was observed except for the rash mentioned later, but it would be good to state this in the manuscript. An option is to even start with a separate section with the experimental design and clinical monitoring that can even include the information currently in lines 178-183 (CBC, fever and weight). 

- Line 180: "...anemia that coincided with peak viremia....". It looks that anemia was more a gradual decline rather than coinciding with peak viremia. This may be related to the frequent sedations and blood collections that may have been close to the limit for this animal?

- Table 1: as this table includes many joints, can the authors clarify, perhaps in the materials and methods, whether when joint tissues were collected, the focus was on mostly joint capsule, rather than bone or cartilage, to test for virus levels?

- Lines 192 to 203: As this is a detailed description of skin pathology, this would fit better later in the manuscript where the rest of the pathology is described. 

- Line 253: the statement "...and variations were absent microscopically where present on sections stained with...". Do the authors mean "macroscopically"?

- Line 276: fix the typo in "innate"

- As listed earlier, Figure 10 that was in the pdf to review is distorted.

- Line 448: As the authors bring up the liver, did they monitor the animals in this study with any serum chemistry panels? If they did, they can mention it in the results. Otherwise no need to test the samples as the likelihood to find something significant is small. 

- Lines 487-488: I suggest to also add longterm pathological changes as a benefit of doing a study with longer duration.

- Line 540: As the results mentioned the the inoculum was administered on both arms, it is good to also mention it here.

- Line 772 : I don't feel the term hematopoietic is the best one to describe the figure, as the data are lymphoid pathology. A similar comment for line 798, but as there bone marrow is included, the header can include both.

- This comment is optional for the authors whether or not to do this, but it looks that the manuscript uses actual animal ID numbers. To avoid getting FOIA requests about these animals, the authors can consider changing the animal numbers. 

-

Reviewer #2: (No Response)

Reviewer #3: (No Response)

**Summary and General Comments**

Reviewer #1: This is a very comprehensive manuscript on the development of a nonhuman primate mode of MAYV. The authors have performed a very thorough analysis based on the available samples. This is a major contribution to the field.

Reviewer #2: Weber and colleagues evaluated the Mayaro virus pathogenesis and immunity in rhesus macaques. The study is good and makes an important contribution to understanding Mayaro disease despite the limitation of 3 NHPs. I highlight the important findings related to crossing the blood-brain barrier for MAYV, which have also been reported to chikungunya, particularly in severe outcomes, including deaths. However, some aspects need to be addressed before publication. 

Major:

1. The manuscript is too extensive and needs to be trimmed. For example, the Introduction section is too long and repetitive. Please revise this section and keep only the essential contextualization for this study. Also, the Result section could be more limited to findings in the study (i.e., only results). Comparison with other studies should be explored in the Discussion section. Please be concise; this will help better understand of study and improve the readership.

2. Please clarify why was selected these MAYV strains. This represents the different genotypes described in the Introduction section. Or was this chose was arbitrary and random? Please explain the rationale. 

3. The titles in the Result section should be redone to summarize the main findings presented in each section (please see also in minor comments). This can call more attention to readers.

4. Consider including in the Discussion section the correlation between CNS and heart infection reported in clinical outcomes and experimental chikungunya virus infection. Please see below some examples of references that can help to contextualize it:

Noval et al. MAVS signaling is required to prevent persistent chikungunya heart infection and chronic vascular tissue inflammation. Nat Commun. 2023 Aug 3;14(1):4668. doi: 10.1038/s41467-023-40047-w. PMID: 37537212; PMCID: PMC10400619.

de Lima STS et al. Fatal Outcome of Chikungunya Virus Infection in Brazil. Clin Infect Dis. 2021 Oct 5;73(7):e2436-e2443. doi: 10.1093/cid/ciaa1038. PMID: 32766829; PMCID: PMC8492446.

5. Immunopathologic could be better explored, including brain and heart infection findings with MAYV (No comment in this current version). Also, I suggest immunoassaying for MAYV in brain and heart tissue to understand the potential replication sites in this tissue since this seems common in severe chikungunya cases. 

6. Please revise the main conclusion for this study for the most relevant findings supported by data (Lines 488-490). In short, the study contributes to understanding the pathogenesis and immunity. However, this impact seems more limited regarding the MAYV transmission, vaccine, monoclonal and antiviral. 

7. Mayaro strain BeAr505411 is not “Buenos Aires 505411” because this strain was isolated in Pará State, Brazil, in 1991 instead of “Buenos Aires, Argentina.” The meaning of “BeAr” is Belém city in Pará State, Brazil, and “Ar” is “Arbovirology section,” where the Evandro Chagas Insitute has performed this viral isolation. Please correct it.

10. Consider a summary scheme to explain the main findings. This could help the reader understand the contribution of this work. 

Figures and tables:

Figure 1. Consider including the experimental design as included in Figure 2.

Figure 2. In B, there is an extra triangle; please remove it. Since there are no body weight changes, consider move D for supplementary. 

Figure 3. This could be moved for supplementary.

Figure 4. Please consider simplifying this figure in the main figures and moving details as supplementary figures. 

Figure 5. This could be moved for supplementary.

Figure 10. Multiple plots overlap. Please revise it.

Table 1. Since the virus was titred by plaque assay, please consider redoing this table as a plot using the titration instead of only positive and negative. 

Table 2. Consider replacing the table with a heatmap format to visualize this data better.

Minors:

Line 58: Replace “Western hemisphere” with “Latin American and Caribbean”

Line 61: Delete “Western”.

Line 61: Replace “Chikungunya” with “chikungunya”.

Line 63: Consider replacing “WHO reference” with “Suhrbier A. et al. Nat Rev Rheumatol. 2019 Oct;15(10):597-611. doi: 10.1038/s41584-019-0276-9. Epub 2019 Sep 3. PMID: 31481759.” 

Line 64: Consider replacing “ECDC reference” with “de Souza WM et al. Lancet Microbe. 2023 May;4(5):e319-e329. doi: 10.1016/S2666-5247(23)00033-2. Epub 2023 Apr 6. PMID: 37031687; PMCID: PMC10281060.

Lines 66-67: Some studies evaluated the cross-reactions between both viruses, such as:

- Webb et al. Sci Rep. 2019 Dec 31;9(1):20399. doi: 10.1038/s41598-019-56551-3. PMID: 31892710; PMCID: PMC6938517.

- Fumagalli et al. J Virol. 2021 Nov 9;95(23):e0112221. doi: 10.1128/JVI.01122-21. Epub 2021 Sep 22. PMID: 34549980; PMCID: PMC8577356.

- Martins et al. Neutralizing Antibodies from Convalescent Chikungunya Virus Patients Can Cross-Neutralize Mayaro and Una Viruses. Am J Trop Med Hyg. 2019 Jun;100(6):1541-1544. doi: 10.4269/ajtmh.18-0756. PMID: 31017081; PMCID: PMC6553899.

Lines 70-72: Most findings related to Aedes mosquitoes being able to transmit MAYV are from experimental infection; please clarify that it is linked to experimental studies. For example: 

Cereghino et al. PLoS Pathog. 2023 Apr 5;19(4):e1010491. doi: 10.1371/journal.ppat.1010491. PMID: 37018377; PMCID: PMC10109513.

Pereira et al. PLoS Negl Trop Dis. 2020 Apr 14;14(4):e0007518. doi: 10.1371/journal.pntd.0007518. PMID: 32287269; PMCID: PMC7182273.

Reviewer #3: (No Response)

PLOS authors have the option to publish the peer review history of their article (what does this mean?). If published, this will include your full peer review and any attached files.

Reviewer #1: No

Reviewer #2: No

Reviewer #3: No
---

## [Decision Letter · Decision Letter 1]

19 Oct 2023

Dear Dr. Streblow,

We are pleased to inform you that your manuscript 'Mayaro virus pathogenesis and immunity in rhesus macaques' has been provisionally accepted for publication in PLOS Neglected Tropical Diseases.

Best regards,

Gregory Gromowski

Academic Editor

Abdallah Samy

Section Editor

---

## [Editor Report · Acceptance letter]

10 Nov 2023

Dear Dr. Streblow,

We are delighted to inform you that your manuscript, "Mayaro virus pathogenesis and immunity in rhesus macaques," has been formally accepted for publication in PLOS Neglected Tropical Diseases.

Best regards,

Shaden Kamhawi

co-Editor-in-Chief

Paul Brindley

co-Editor-in-Chief
